# VARIANCE-REDUCED REINFORCEMENT LEARNING FOR LARGE REASONING MODELS VIA JAMES-STEIN BASELINES

## ABSTRACT

Reinforcement Learning with Verifiable Rewards (RLVR) is becoming an impactful paradigm in large reasoning model (LRM) post-training. To stabilize training, control variates (baselines) are commonly introduced, canonically chosen to approximate the value function. Popular approaches such as RLOO and GRPO estimate baselines with per-prompt empirical averages of generated response, which can exhibit high variance under limited rollout budgets. Recognizing that value functions must be estimated simultaneously across all prompts in a batch, we propose a James–Stein estimator as the baseline. This approach leverages statistical shrinkage to reduce the mean squared error in the overall value function estimation, without additional computational overhead while maintaining the unbiasedness of the policy gradient estimator. We provide theoretical justification for James-Stein baselines and validate it empirically. Across diverse models, tasks, and rollout budgets, our approach consistently outperforms existing baselines, demonstrating robust variance reduction and improved training stability.

## 1 INTRODUCTION

Recent large reasoning models such as *OpenAI-o1* (OpenAI, 2024) and *DeepSeek-R1* (Guo et al., 2025) have demonstrated impressive reasoning capabilities, underscoring the effectiveness of reinforcement learning (RL) techniques for model post-training. A particularly impactful paradigm for fine-tuning reasoning models is **Reinforcement Learning with Verifiable Rewards (RLVR)**, where models are optimized using sparse, rule-based scalar rewards explicitly indicating the correctness of the model's final answer. RLVR-style training has shown substantial promise for tasks that require explicit and verifiable logic, such as mathematical or logical reasoning.

A common approach to RLVR is applying policy gradient methods such as REINFORCE (Williams, 1992) or more modern variants such as GRPO (Shao et al., 2024), DAPO (Yu et al., 2025) and CISPO (MiniMax, 2025), where the model is optimized to maximize expected rewards through stochastic gradient estimates. A well-known challenge in policy gradient methods is the high variance of these gradient estimators (Sutton and Barto, 2018), which can hinder stable training. To mitigate this, RL methods introduce a baseline—known in statics as a control variate—which is a state-dependent shift in the rewards that reduces variance in the gradient without introducing bias (Sutton et al., 1998). The canonical choice is the value function, defined as the expected return from the state under consideration (or the initial state in RLVR) under the current policy. Despite being a heuristic rather than the theoretically optimal baseline, it is widely adopted because it is straightforward to approximate while providing substantial variance reduction.

Typically, the value function itself is unknown and it must be estimated. There are two broad classes of value function estimators. *Classical RL approaches* introduce an auxiliary neural network to approximate the value function (Barto et al., 1989; Mnih et al., 2016; Haarnoja et al., 2018; Schulman et al., 2015a; 2017). When carefully tuned, this strategy can be effective, but it comes with significant practical challenges: increased hyperparameter sensitivity, added engineering complexity, and the cost of training and maintaining an additional network for variance reduction. In contrast, *recent methods for reasoning models*—including GRPO (Shao et al., 2024), RLOO (Ahmadian et al., 2024), ReMax (Li et al., 2023), REINFORCE++ (Hu, 2025), DAPO (Yu et al., 2025) and CISPO (MiniMax, 2025)—forgo explicit value function approximation. Instead, they construct baselines directly from

Monte Carlo returns, typically using per-prompt empirical averages of generated responses. This avoids the overhead of a separate network and yields unbiased (or nearly unbiased) estimates, making it attractive for large reasoning models.

Monte Carlo–based baselines are simple and unbiased. Yet with small rollout budgets, their empirical averages often exhibit high variance, as in practice only 2–8 rollouts are typically used (Shao et al., 2024; Zhang and Zuo, 2025; Phan et al., 2025), leaving room for improved alternatives. In this work, we revisit the statistical problem of estimating value functions across prompts. Although the per-prompt sample mean is an unbiased estimator, it treats each prompt independently. By recognizing that value functions must be estimated **simultaneously** across all prompts in a batch, we can construct an estimator for the value function—and thus the gradient—with strictly lower mean squared error (MSE). In particular, we propose a new baseline estimator inspired by the classical James–Stein shrinkage principle (James et al., 1961; Stein et al., 1956). This estimator reduces variance by trading a small amount of bias in the baseline for improved overall efficiency. Crucially, despite using a biased baseline, the **resulting policy gradient estimator remains unbiased** and enjoys provable variance reduction under standard assumptions.

Our proposed baseline introduces **no additional hyperparameters**, making it a simple drop-in replacement for existing critic-free RL methods. It relies solely on frequentist principles, without requiring assumptions about task difficulty, training data distributions, or model architectures. Importantly, the James-Stein shrinkage baseline can be computed with **negligible computational overhead**. Extensive experiments across diverse models, tasks, and rollout settings demonstrate that the James-Stein shrinkage baseline estimator consistently outperforms other common baselines in variance reduction. Furthermore, we observe a significant decrease in policy gradient variance, aligning with our theoretical predictions.

## 2 PRELIMINARIES

Let $\pi_\theta$ denote the language model parameterized by $\theta$. Given a prompt $x$ sampled uniformly from an unknown prompt distribution $\mathcal{D}$, the language model outputs a response $y$ with probability $\pi_\theta(y \mid x)$ and receives reward $r(x, y)$. We consider the verifiable reward setting, i.e., $r$ is a deterministic and known function. For instance, in math problem-solving tasks, the reward is 1 if the response gives the correct answer, and 0 otherwise. The reinforcement learning objective is to maximize the expected reward

$$J(\theta) := \mathbb{E}_{x\sim\mathcal{D}, y\sim\pi_\theta(\cdot|x)}[r(x, y)].$$

The REINFORCE algorithm (Williams, 1992) derives the policy gradient as

$$\nabla_\theta J(\theta) = \mathbb{E}_{x\sim\mathcal{D}, y\sim\pi_\theta(\cdot|x)}[r(x, y)\nabla_\theta \log \pi_\theta(y \mid x)],$$

so that the gradient can be estimated with one online sample:

$$g^{\mathrm{vanilla}}(x, y; \theta) = r(x, y)\nabla_\theta \log \pi_\theta(y \mid x).$$

A scalar, prompt-dependent baseline $b(x) \in \mathbb{R}$ can be added to further reduce the gradient variance while keeping the gradient unbiased:

$$g^{\mathrm{baseline}}(x, y; \theta) := (r(x, y) - b(x))\nabla_\theta \log \pi_\theta(y \mid x). \tag{1}$$

In general, multiple online samples can be used to further reduce variance. At each RL step, we sample $n$ prompts $x_1, x_2, \cdots, x_n$ i.i.d. from $\mathcal{D}$. For each prompt $x_i$, the language model $\pi_\theta$ generates $m$ responses $y_i^1, y_i^2, \cdots, y_i^m$ independently from $\pi_\theta(\cdot \mid x_i)$ and observes the rewards $r_i^j := r(x_i, y_i^j)$ $(1 \le j \le m)$. Let $\mathbf{x} = (x_1, x_2, \cdots, x_n)$, $\mathbf{y}_i = (y_i^1, y_i^2, \cdots, y_i^m)$ $(1 \le i \le n)$, $\mathbf{Y} = (\mathbf{y}_1, \mathbf{y}_2, \cdots, \mathbf{y}_n)$. The previously described distribution is the default distribution of $\mathbf{x}$ and $\mathbf{Y}$ unless explicitly stated otherwise. The policy gradient can then be estimated by

$$g(\mathbf{x}, \mathbf{Y}; \theta) := \frac{1}{n}\sum_{i=1}^{n}\frac{1}{m}\sum_{j=1}^{m}(r_i^j - b_i^j)\nabla_\theta \log \pi_\theta(y_i^j|x_i). \tag{2}$$

Here, $b_i^j$ denotes the baseline associated with sample $(x_i, y_i^j)$. In practice, baselines are usually prompt-dependent (i.e., $b(x_i)$) and shared across responses, but we present the more general notation

here because recent leave-one-out estimators such as RLOO (Ahmadian et al., 2024) adopt slightly different baselines per reward to ensure unbiasedness[1], an approach that we also follow.

As shown in Proposition 1, Equation (2) is an unbiased estimate of the policy gradient $\nabla_\theta J(\theta)$, as long as $b_i^j$ and $r_i^j$ are independent for all $1 \leq i \leq n, 1 \leq j \leq m$.

**Proposition 1 (Unbiasedness)** *Suppose $b_i^j$ is independent of $y_i^j$ for all $1 \leq i \leq n$ and $1 \leq j \leq m$. Then $g(\mathbf{x}, \mathbf{Y}; \theta)$ is unbiased. That is,*

$$\mathbb{E}[g(\mathbf{x}, \mathbf{Y}; \theta)] = \nabla_\theta J(\theta).$$

We defer the proof to Section D.1. REINFORCE with baseline is a special form of Equation (2) when $n = m = 1$.

Beyond this basic formulation, several practical algorithms have been developed to improve training stability. PPO (Proximal Policy Optimization) (Schulman et al., 2017) is widely used in RLHF pipelines for language models due to its clipping objective, which prevents overly large policy updates. It is defined as by the following updates, where we leave the advantage estimator $A_{i,j,t}$ to be specified.

$$\mathcal{J}_{\text{PPO}}(\theta) = \frac{1}{n} \sum_{i=1}^{n} \frac{1}{m} \sum_{j=1}^{m} \frac{1}{|y_i^j|} \sum_{t=1}^{|y_i^j|} \min\Big(\rho_{i,j,t}(\theta) \, A_{i,j,t}, \, \text{clip}(\rho_{i,j,t}(\theta), 1 - \epsilon, 1 + \epsilon) \, A_{i,j,t}\Big)$$

$$\rho_{i,j,t}(\theta) = \frac{\pi_\theta(y_{i,t}^j \mid x_i, y_{i,<t}^j)}{\pi_{\theta_{\text{old}}}(y_{i,t}^j \mid x_i, y_{i,<t}^j)}$$

More recently, reasoning-model-specific variants such as GRPO (Ahmadian et al., 2024) have proposed Z-normalized advantage estimators which eliminate the use of additional networks for the advantage estimation. (Here $\delta > 0$ is a small constant to avoid division by 0.)

$$\mu_i = \frac{1}{m} \sum_{j=1}^{m} r_i^j, \qquad \sigma_i = \sqrt{\frac{1}{m-1} \sum_{j=1}^{m} \Big(r_i^j - \mu_i\Big)^2}, \qquad A_{i,j}^{\text{GRPO}} = \frac{r_i^j - \mu_i}{\sigma_i + \delta}.$$

In this work, we keep the standard RLVR setting with a sparse, terminal reward $r(x, y)$ and focus on improving the trajectory-level baseline for variance reduction. While centering the reward by the value function is standard RL practice (Sutton and Barto, 2018), the division by the prompt standard deviation is a key distinguishing feature of GRPO. However, recent work has established that the division by the standard deviation biases the objective function (Liu et al., 2025) without necessarily increasing empirical performance and it can thus be omitted (Khatri et al., 2025).

## 3 DERIVATION OF THE METHOD

### 3.1 FROM POLICY GRADIENT VARIANCE TO VALUE FUNCTION ESTIMATORS

The purpose of a reinforcement learning baseline is to act as a control variate and thereby reduce the variance of the policy gradient estimator. The baseline $b$ should be chosen so as to minimize the variance of the gradient estimator in eq. (2). Since $g(\mathbf{x}, \mathbf{Y}; \theta)$ is vector-valued, its variance is naturally represented by the covariance matrix $\text{Var}[g(\mathbf{x}, \mathbf{Y}; \theta)]$. A common scalar summary is the trace of this matrix, i.e., the sum of coordinate-wise variances, which is equivalent to the mean-squared error of the estimator:

$$\text{Var}[g] := \text{Tr}\big(\text{Var}[g(\mathbf{x}, \mathbf{Y}; \theta)]\big) = \mathbb{E}\left[\|g(\mathbf{x}, \mathbf{Y}; \theta) - \nabla_\theta J(\theta)\|_2^2\right]. \tag{3}$$

Minimizing this quantity by means of an appropriate baseline $b$ corresponds to constructing more efficient gradient estimators, which is the central focus of this work.

In general, the baseline that minimizes its variance is a complicated function of both the prompt and the response. The variance-minimizing baseline for a given prompt $x$ is known to be dependent on

---

[1]In these estimators, the baseline for each reward is computed by leaving out that reward itself (e.g., using the average of the other rewards for the same prompt). This ensures the baseline remains independent of the reward it is paired with, which is necessary for unbiasedness.

the squared norm of the score function $\nabla_\theta \log \pi_\theta(y \mid x)$, which is typically expensive to compute or approximate in practice (Greensmith et al., 2004).

For this reason, a standard simplification in the literature—often made implicitly—is to ignore the dependence on the score function and directly find a baseline function $b(x)$ that minimizes the Mean Square Error of the baseline estimator with respect to the observed rewards. In other words, theoretically one would choose a baseline $b$ to minimize the population-level mean square error (for every state $x$)

$$\mu(x) = \arg\min_b \mathbb{E}[(r(x,y) - b(x))^2], \tag{4}$$

which minimizes eq. (3) when the score function is ignored. In other words, the optimal baseline is the value function when that is realizable by the function class of the baseline, which is the *Bayes-optimal predictor* of the reward under the current policy.

$$\mu(x) := \mathbb{E}_{y \sim \pi_\theta(\cdot \mid x)}[r(x,y)], \quad \forall x.$$

Although this choice is not strictly the optimal variance-minimizing baseline for the full policy gradient (which depends on the score function norm), it is a close approximation which enjoys a clear statistical interpretation. Therefore, the value-function baseline has become the standard foundation for variance reduction in reinforcement learning (Sutton and Barto (2018)) . It directly motivates both classical actor–critic methods (e.g., A3C (Mnih et al., 2016), SAC (Haarnoja et al., 2018), TRPO (Schulman et al., 2015a), PPO (Schulman et al., 2017)) and more recent critic-free methods tailored for reasoning models (e.g., ReMax (Li et al., 2023), RLOO (Ahmadian et al., 2024), GRPO (Shao et al., 2024), REINFORCE++ (Hu, 2025)), and is also the starting point for our development to follow.

In practice, the value function is unknown. Simple algebra shows that minimizing eq. (4) leads to **minimizing the mean squared error of the baseline** $b(x)$ **with respect to the value function** $\mu(x)$, which is the starting point for our development to follow.

$$\begin{aligned}
&\mathbb{E}[(r(x,y) - b(x))^2] \\
=\ &\mathbb{E}[(r(x,y) - \mu(x) - b(x) + \mu(x))^2] \\
=\ &\mathbb{E}[(r(x,y) - \mu(x))^2] - 2\mathbb{E}[(r(x,y) - \mu(x))(b(x) - \mu(x))] + \mathbb{E}[(b(x) - \mu(x))^2] \\
=\ &\mathrm{Var}[r(x,y)] + \mathbb{E}[(b(x) - \mu(x))^2]. \tag{5}
\end{aligned}$$

## 3.2 A Bias-Variance Tradeoff for Baselines in RLVR

In RLVR, most critic-free methods rely solely on *prompt-level* samples—i.e., responses to the same prompt—to construct the baseline. Yet practical policy-gradient updates require value-function estimates for a *batch* of prompts. This means that the empirical optimization program corresponding to eq. (5) is $\frac{1}{mn} \sum_{i=1}^n \sum_{j=1}^m \mathbb{E}_{\mathbf{Y}}[(b_i^j - \mu_i)^2]$. Consider the standard bias–variance decomposition of the baseline mean-squared error. [2]

$$\frac{1}{mn} \sum_{i=1}^n \sum_{j=1}^m \mathbb{E}\big[(b_i^j - \mu_i)^2\big] \tag{6}$$

$$= \frac{1}{mn} \sum_{1 \leq i \leq n, 1 \leq j \leq m} \mathbb{E}[(b_i^j - \mathbb{E}[b_i^j] + \mathbb{E}[b_i^j] - \mu_i)^2]$$

$$= \frac{1}{mn} \sum_{1 \leq i \leq n, 1 \leq j \leq m} \left\{ \mathbb{E}[(b_i^j - \mathbb{E}[b_i^j])^2] + 2\mathbb{E}[(b_i^j - \mathbb{E}[b_i^j])](\mathbb{E}[b_i^j] - \mu_i) + (\mathbb{E}[b_i^j] - \mu_i)^2 \right\}$$

$$= \underbrace{\frac{1}{mn} \sum_{1 \leq i \leq n, 1 \leq j \leq m} \mathbb{E}[(b_i^j - \mathbb{E}[b_i^j])^2]}_{\text{Variance}} + \underbrace{\frac{1}{mn} \sum_{1 \leq i \leq n, 1 \leq j \leq m} (\mathbb{E}[b_i^j] - \mu_i)^2}_{\text{Bias}^2}.$$

---

[2]Throughout Section 3.2, analyze a single RL step conditional on the sampled prompt batch $\mathbf{X}$. For a realization $\mathbf{x}$, all expectations/variances are over rollouts only: $\mathbb{E}_{\mathbf{Y}}[\cdot] := \mathbb{E}[\cdot \mid \mathbf{X} = \mathbf{x}]$ and $\mathrm{Var}_{\mathbf{Y}}(\cdot) := \mathrm{Var}(\cdot \mid \mathbf{X} = \mathbf{x})$. We drop the subscript when unambiguous.

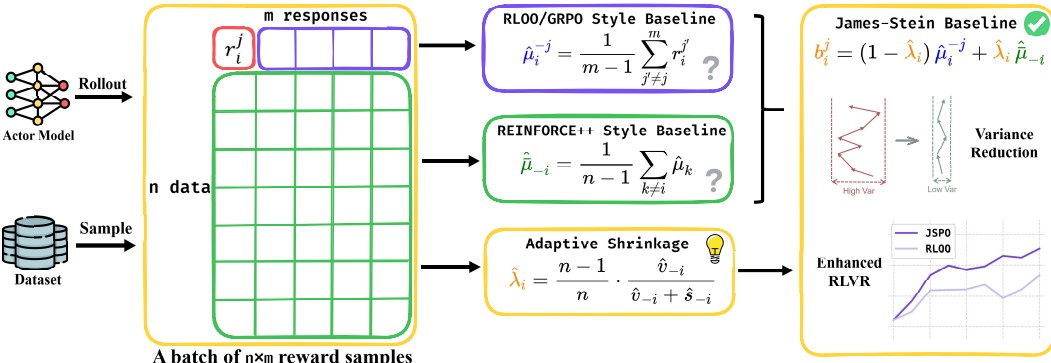

Figure 1: **Overview of using the James-Stein Shrinkage Baseline in RLVR of large reasoning models.** Consider a step in RLVR with **n** question prompts, each generating **m** responses. For every response, our method computes a leave-one-out prompt-level reward mean $\widehat{\mu}_i^{-j}$ and a leave-one-out batch-level reward mean $\widehat{\mu}_{-i}$. It then estimates an optimal shrinkage coefficient $\widehat{\lambda}_i$ from reward-sample statistics. These components are combined to produce a variance-reduced baseline $b_i^j$. By lowering the variance in policy-gradient estimation, the JS baseline enables more effective reinforcement learning for large reasoning models.

Here expectations are taken over the responses $\mathbf{Y}$, and $\mu_i = \mathbb{E}_{y \sim \pi_\theta(\cdot|x_i)}[r(x_i, y)]$ denotes the true value function for prompt $x_i$. For exposition, we temporarily relax the unbiased-gradient requirement that $b_i^j$ be independent of $y_i^j$.

The decomposition above highlights a bias–variance trade-off, with different baselines exhibiting distinct behaviors. For example, choosing the *prompt-level mean reward* $\widehat{\mu}_i = \frac{1}{m} \sum_{j=1}^m r_i^j$ for each baseline, as in RLOO (Ahmadian et al., 2024) and GRPO (Guo et al., 2025), minimizes the bias term. However, this choice does not efficiently reduce variance because it ignores informative cross-prompt structure. At the other extreme, one may use the *global batch mean* $\widehat{\widehat{\mu}} = \frac{1}{nm} \sum_{i=1}^n \sum_{j=1}^m r_i^j$, an idea adopted in recent works such as REINFORCE++ (Hu, 2025) and SPO (Xu and Ding, 2025). This baseline often achieves low variance but incurs substantial bias when prompt-specific means differ.

In such setting it is useful to recall **Stein's paradox** (Stein et al., 1956): when estimating multiple means jointly, the empirical mean is provably suboptimal. Shrinkage estimators that pool information across tasks—even when those tasks are independent—can strictly reduce total MSE. A natural approach is therefore to interpolate between the two extremes by shrinking each prompt-level mean toward the global batch mean—the core idea behind the James–Stein (JS) estimator (James et al., 1961; Stein et al., 1956):

$$b_i^{j,\mathrm{JS1}} = (1 - \lambda)\,\widehat{\mu}_i + \lambda\,\widehat{\widehat{\mu}}. \tag{7}$$

The shrinkage coefficient $\lambda \in [0, 1]$ balances variance reduction and bias. The optimal $\lambda$ admits a closed-form estimate from data (proof in Appendix D.2):

> **Proposition 2** *Let* $v = \frac{1}{nm} \sum_{i=1}^n \sigma_i^2$, *where* $\sigma_i^2 = \mathrm{Var}[r(x_i, y)]$. *Let* $\bar{\mu} = \frac{1}{n} \sum_{i=1}^n \mu_i$ *and* $s = \frac{1}{n-1} \sum_{i=1}^n (\mu_i - \bar{\mu})^2$. *Then the minimizer of the relaxed MSE is*
>
> $$\lambda^\star = \frac{v}{s+v}.$$

Here $v$ captures the average per-prompt variance, while $s$ measures the dispersion of true value functions across prompts. When prompts are similar (small $s$), stronger shrinkage is preferred; when prompts are heterogeneous (large $s$), the estimator leans more on local means. Since $0 < \lambda^\star < 1$ whenever $v > 0$ and $s > 0$ (here we don't consider the unlikely case where all prompts in the batch and all responses have reward 1 (or 0) for simplicity of exposition), both the prompt-level mean reward and the global batch mean reward are strictly suboptimal for variance reduction. Instead, the

baseline that interpolates them together as Equation (7) is the optimal balance between *batch-level* and *prompt-level* information, which outperforms both empirical means under objective 6.

## 3.3 RLVR with James–Stein Baseline

The analysis in the prior section suggests that a James–Stein baseline with lower MSE can substantially reduce policy gradient variance. However, a critical additional requirement in policy gradient is *unbiasedness*. The naive shrinkage baseline in Equation (7) cannot be used directly, since it is correlated with the rewards $r_i^j$ that appear in the gradient estimator, and thus would introduce bias. In addition, we need to consider the general case that prompts are sampled from $\mathcal{D}$ instead of being fixed.

To guarantee independence between the baseline and each individual reward, we adopt a two-level leave-one-out construction in the spirit of RLOO. For each prompt $x_i$ and response $y_i^j$, define

$$\widehat{\mu}_i^{-j} := \frac{1}{m-1} \sum_{j' \neq j} r_i^{j'} \qquad \text{(leave-one-out prompt-level average)} \qquad (8)$$

$$\widehat{\widehat{\mu}}_{-i} := \frac{1}{n-1} \sum_{k \neq i} \widehat{\mu}_k \qquad \text{(leave-one-out global batch average)} \qquad (9)$$

$$b_i^{j,\text{JS2}} := (1 - \lambda_i^j)\, \widehat{\mu}_i^{-j} + \lambda_i^j\, \widehat{\widehat{\mu}}_{-i}. \qquad (10)$$

Compared to the naive baseline, Equation (10) replaces both the prompt-level and batch-level means with leave-one-out counterparts, ensuring that $b_i^{j,\text{JS2}}$ is independent of the held-out reward $r_i^j$. Furthermore, we allow the shrinkage coefficient to vary by sample, i.e., each $b_i^j$ uses its own $\lambda_i^j$, which can also be chosen independently of $r_i^j$. With these modifications, the resulting estimator yields an unbiased policy gradient.

The optimal shrinkage coefficient has essentially the same form as in the naive James–Stein estimator. Define $\mu(x) := \mathbb{E}_{y \sim \pi(\cdot|x)}[r(x, y)]$ and $\sigma^2(x) := \text{Var}_{y \sim \pi(\cdot|x)}[r(x, y)]$. The following theorem provides its precise expression.

> **Theorem 1** *Let $v_2 = \frac{1}{m-1}\mathbb{E}_{x \sim \mathcal{D}}[\sigma^2(x)]$ and $s_2 = \text{Var}_{x \sim \mathcal{D}}[\mu(x)]$. Then the optimal James–Stein coefficient for Equation (6) is the same across all prompts $i$ and samples $j$, and is given by*
>
> $$(\lambda_i^j)^* = \frac{n-1}{n} \cdot \frac{v_2}{s_2 + v_2}. \qquad (11)$$

Here $v_2 = \frac{1}{m-1}\mathbb{E}_{x \sim \mathcal{D}}[\sigma^2(x)]$ measures the *expected variance* of the leave-one-out local estimator $\widehat{\mu}_i^{-j}$, while $s_2 = \text{Var}_{x \sim \mathcal{D}}[\mu(x)]$ quantifies the variability of the true value functions across prompts. In Appendix D.3, we prove Theorem 1 under mild assumptions, without assuming any particular parametric form for the conditional reward distribution $r(x, Y) \mid x$.

### Intuition of Theorem

- When $v_2 \gg s_2$, the per-prompt estimates are highly noisy, so stronger shrinkage toward the global average is optimal. This typically occurs when only few rollouts per prompt (e.g., $m = 2$) are available.

- When $s_2 \gg v_2$, the value functions vary substantially across prompts, so shrinkage toward the global mean would introduce excessive bias. In this regime, the optimal $\lambda$ is close to zero, approaching the leave-one-out prompt-level mean.

Thus, $(\lambda_i^j)^*$ achieves the optimal trade-off between variance reduction and bias control, while preserving the independence condition required for unbiased policy gradients.

**Implementation details** In practice, we first estimate $\widehat{v}_{-i}$ and $\widehat{s}_{-i}$ from statistics among batch leave-one-out reward samples:

$$\widehat{v}_{-i} = \frac{1}{n-1} \sum_{k \neq i} \left( \frac{1}{m(m-1)} \sum_{j=1}^{m} (r_k^j - \widehat{\mu}_k)^2 \right), \quad \widehat{s}_{-i} = \frac{1}{n-1} \sum_{k \neq i} (\widehat{\mu}_k - \widehat{\overline{\mu}}_{-i})^2. \quad (12)$$

Given these plug-in estimates, the per-prompt shrinkage coefficient becomes

$$\widehat{\lambda}_i = \frac{n-1}{n} \cdot \frac{\widehat{v}_{-i}}{\widehat{v}_{-i} + \widehat{s}_{-i}}. \quad (13)$$

Finally, combining the local averages (Equation (8)) with global ones (Equation (9)) with the estimated $\widehat{\lambda}_i$ in Equation (13) yields the *James–Stein baseline*:

$$b_i^j = (1 - \widehat{\lambda}_i) \, \widehat{\mu}_i^{-j} + \widehat{\lambda}_i \, \widehat{\overline{\mu}}_{-i}. \quad (14)$$

It is worth noting that $b_i^j$ here does not depend on the response $y_i^j$, so it satisfies the condition of Proposition 1, and thus the resulting gradient is **unbiased**.

## 4 EXPERIMENTAL ANALYSIS

In this section, we empirically evaluate the effectiveness of proposed James-Stein Baseline in reinforcement finetuning of large reasoning models. We show that our method improves the efficacy of reinforcement learning by reducing the variance of policy gradient. We adopt the GRPO algorithm (Shao et al., 2024) without advantage normalization as recommended by a concurrent recent large scale empirical investigation (Khatri et al., 2025), and with leave-one-out reward centering; this algorithm is also known as RLOO (Ahmadian et al., 2024).

### 4.1 MATHEMATICAL REASONING

We first evaluate JS baseline on mathematical reasoning tasks. In this section, we choose Qwen2.5-Math-1.5B (Team, 2024), Qwen2.5-Math-7B (Team, 2024) and Qwen3-4B-Base (Yang et al., 2025) as base models, and our training dataset includes DAPO17k (Yu et al., 2025) and MATH12k (Hendrycks et al., 2021). During training, we set 64 questions per batch and 4 rollouts per question. For Qwen2.5 math models, we set max number of tokens to 2048 and evaluate on three commonly used benchmarks: MATH500 (Hendrycks et al., 2021), OlympiadBench (He et al., 2024) and AMC23 (math ai, 2025). For Qwen3-4B-Base model, we expand the max number of tokens to 3072, increase the clip ratio, and adopt length-dependent loss aggregation technique following DAPO (Yu et al., 2025). To ensure fairness of comparison, data loading sequence and random seeds are all fixed.

Figure 2 shows the accuracy of math reasoning on Qwen2.5 math models, evaluated by average Pass@1 in 16 samples, and Figure 3 illustrates the training reward curve and test accuracy of Qwen3-4B-Base model. We can see that compared with RLOO baseline(Ahmadian et al., 2024), which only uses leave-one-out average reward in a single question as baseline and without shrinkage to batch mean, JS baseline illustrates substantial improvement in different models and benchmarks, with a 1.1% ~ 4.3% gain in accuracy and a significantly faster reward improvement during training. In Appendix C.2, we include additional experiment details and more results.

### 4.2 LOGIC PUZZLE REASONING

We then extend our James-Stein baseline to various logic puzzle reasoning tasks and models. Following previous work (Pan et al., 2025; Chen et al., 2025; Stojanovski et al., 2025), we adopt three settings that are suitable for reinforcement finetuning in terms of model capability and task difficulty: Qwen2.5-7B-Instruct (Team, 2024) model on Knights-and-Knaves (KnK) (Stojanovski et al., 2025), Qwen2.5-3B (Team, 2024) model on Countdown (Gandhi et al., 2024; Pan et al., 2025) and Ministral-8B-Instruct (Jiang et al., 2024) model on Maze (Chen et al., 2025). For each model, we train for at least 200 steps and evaluate on a test set of at least 200 problems within the same distribution of the training set, without overlapping with training data. Apart from the experiments above, we additionaly train Qwen2.5-1.5B-Instruct (Team, 2024) on KnK dataset for as much as 1000 steps in a smaller learning rate to further validate the effectiveness of our method. We provide a detailed illustration of each puzzle, additional information and experimental results in Appendix C.3.

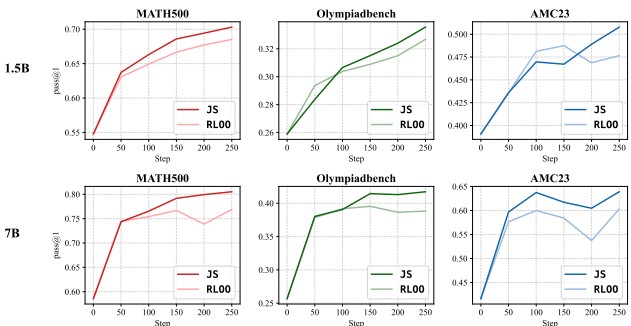

Figure 2: Comparison of JS shrinkage baseline with RLOO (Ahmadian et al., 2024) baseline on Qwen2.5 math models trained on DAPO17k and MATH12k datasets. JS baseline significantly outperforms RLOO across different models and benchmarks.

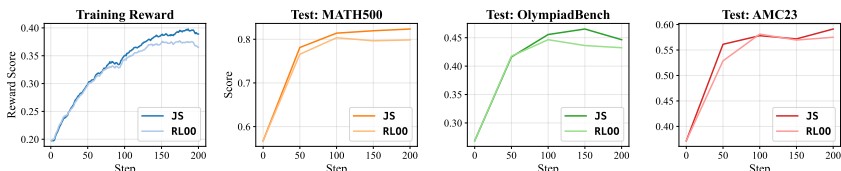

Figure 3: Comparison of training reward and test accuracy between JS baseline and RLOO on Qwen3-4B-Base model trained on DAPO17k dataset.

Figure 4 plots the curves of average test scores with respect to steps in different logic puzzles, and Figure 5 demonstrates the training reward and test accuracy on Qwen2.5-1.5B-Instruct model in the long run. Compared with RLOO baseline, JS baseline shows substantial improvement (2.3% ~ 15.2%) in terms of model capability on all three tasks. The results demonstrate that JS baseline implements more stable and effective parameter updates during the training of reasoning LLMs.

## 4.3 COMPARISON WITH OTHER VARIANCE REDUCTION BASELINES

We then explore the performance of JS baseline under different number of rollouts and systematically compare with other variance reduction baselines. For computation efficiency, we train Qwen2.5-0.5B-Instruct (Team, 2024) model on GSM8k (Cobbe et al., 2021) dataset for 500 steps. For each RL step, we sample 64 questions, and vary the number of rollouts (i.e. number of generations per question) among 2,4,8. We experiment on different critic-free RLVR baselines, including GRPO baseline (Shao et al., 2024), RLOO baseline (Ahmadian et al., 2024), ReMax baseline (Li et al., 2023), REINFORCE++ baseline (Hu, 2025), batch-level leave one out (BLOO) and JS baseline. BLOO means computing the baseline by the average of rewards within the batch with the current prompt left out, i.e., it uses $\widehat{\mu}_{-i}$ in Equation (14). For each setting, we iterate over 5 random seeds and report the average final accuracy on the GSM8k test split.

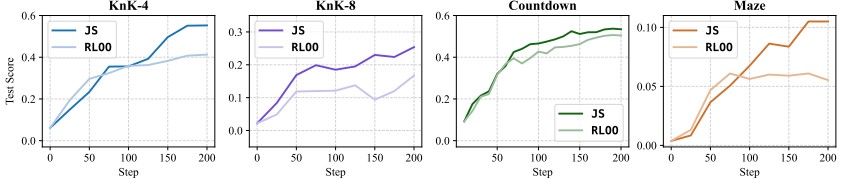

Figure 4: Comparison of average scores on test set between JS baseline and RLOO on Logic Puzzle Reasoning Tasks. JS baseline outperforms RLOO across various tasks and models. Note that the number after Knights-and-Knaves (KnK) datasets denotes the quantity of people in the puzzle. Larger number suggests higher difficulty.

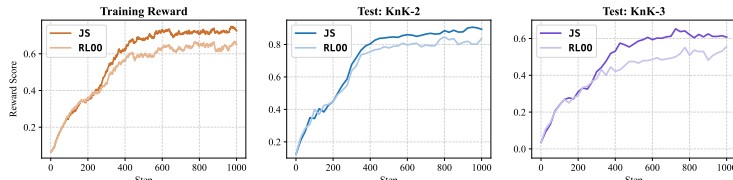

Figure 5: Comparison on training reward (running average) and test accuracy between JS baseline with RLOO on Qwen2.5-1.5B-Instruct model and KnK dataset.

As shown in Table 1, using the JS baseline consistently achieves the highest evaluation scores across all rollout settings, whereas competing baselines only excel under specific conditions. RLOO and GRPO perform best with 8 rollouts, where prompt-level reward averaging becomes accurate, while REINFORCE++ and BLOO fare relatively better with only 2 rollouts, since GRPO and RLOO suffer from higher variance and batch-level averaging provides more stability. ReMax++ delivers modest performance across all settings, consistent with the limitations of its greedy decoding design.

Table 1: **Final test accuracy (%) across various number of rollouts.** Best are in bold, second-best with *, third-best with †.

| Baseline | 2 Gen | 4 Gen | 8 Gen |
|---|---|---|---|
| ReMax | 54.70† | 56.47* | 57.76 |
| REINFORCE++ | 54.82* | 55.27 | 57.30 |
| GRPO | 53.68 | 56.24 | 58.28† |
| BLOO | 54.19 | 56.06 | 57.22 |
| RLOO | 54.49 | 56.34† | 58.31* |
| JS | **55.22** | **57.33** | **58.93** |

### 4.4 ERROR REDUCTION IN VALUE FUNCTION MSE

Moreover, we estimate the value function of each question be monte-carlo sampling. For each question, we generate 32 trajectories and compute the average reward score as an accurate estimation. After that, we sample another batch of responses under fixed smaller numbers of rollouts, and computed the mean squared error between different baselines and estimated value in a batch of 64 questions, same as the training setting. The MSE metric for RLOO, REINFORCE++, ReMax and JS baseline throughout training are shown as Figure 6. Under different rollout budgets, the JS baseline consistently shows smallest deviation with the value score compared with RLOO, REINFORCE++ and ReMax. Overall, these results support our theoretical derivations and highlight the advantage of the James–Stein estimator for reinforcement finetuning under varying rollout counts.

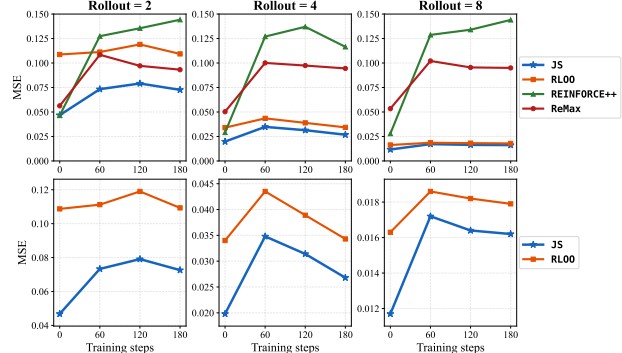

Figure 6: MSE between value score and estimated baseline under different rollout budgets during training. The results are based on the average of 20 randomly selected batches from DAPO17k, and weights from the experiments on Qwen3-4B-Base. With James-Stein baseline shrinkage, the estimation is consistently closer to value score. For 2 rollouts, 4 rollouts, 8 rollouts, mean squared error for JS baseline estimator are 39.4%, 25.1% and 13.4% lower than RLOO estimator, respectively.

### 4.5 ANALYSIS OF TRAINING DYNAMICS

We focus on providing insights into two key training dynamics that reveal the advantage of using the JS baseline: namely, the *Adaptive Shrinkage* of the James–Stein coefficient and the *Reduced Variance* of the policy gradient. We provide more results illustrating these two effects in Appendix C.6.

**Adaptive Shrinkage.** The shrinkage coefficient $\widehat{\lambda}_i$ in Equation (14) is central to JS baseline. Figure 7 reports its average value during training across different rollout counts, revealing two key trends: (i) with more rollouts, $\widehat{\lambda}_i$ decreases, since intra-prompt estimates become more reliable and JS baseline

naturally approaches the RLOO baseline; and (ii) $\widehat{\lambda}_i$ decays over training, as RLVR drives the policy toward greater determinism, reducing the usefulness of cross-prompt references. Together, these behaviors constitute an adaptive shrinkage mechanism that adjusts with both rollout number and training progress, explaining why JS baseline consistently outperforms RLOO and BLOO across all rollout settings in Section 4.3.

**Reduced Variance.** The variance of the policy gradient in Equation (3) is the key metric that reflects the stability of the reinforcement finetuning. To track the gradient variance, we need to build an unbiased estimator of $\mathrm{Var}(g)$ using the observable gradients during training. Following McCandlish et al. (2018), we collect $m$ micro-batches of gradients $g_i$ $(i = 1, \ldots, m)$ in one training step, then an unbiased estimator for $\mathrm{Var}(g)$ becomes

$$\widehat{\mathrm{Var}(g)} = \frac{1}{m} \cdot \frac{1}{m-1} \sum_{i=1}^{m} \|g_i - \bar{g}\|^2 = \frac{1}{m} \cdot \frac{1}{m-1} \left( \sum_{i=1}^{m} \|g_i\|^2 - \frac{1}{m} \left\| \sum_{i=1}^{m} g_i \right\|^2 \right), \quad \bar{g} = \frac{1}{m} \sum_{i=1}^{m} g_i. \tag{15}$$

We incorporate this estimator during training when GPU memory permits additional gradient storage. Further implementation details are provided in Section C.5.

Figure 8 reports the running average of gradient variance across different experiments and models. Compared to RLOO, training with JS baseline reduces gradient variance by 11.2%, 17.4%, 31.6% and 67.1%, respectively. The JS baseline mitigates this issue by consistently reducing variance across rollout counts, models, and tasks, thereby enabling more stable and effective reinforcement learning for reasoning LLMs.

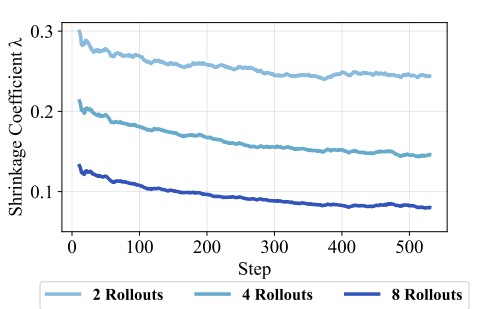

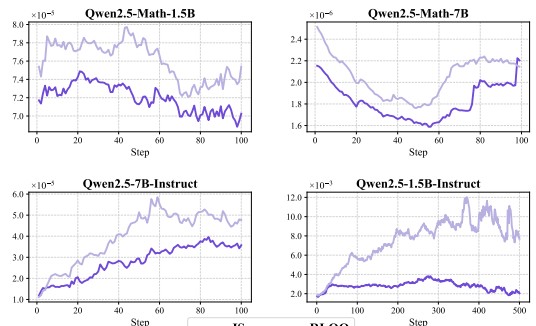

Figure 7: **Moving average shrinkage coefficient.** $\widehat{\lambda}_i$ during training at different numbers of rollouts.

Figure 8: **Estimated variance of policy gradient during training in different models.** With James–Stein baseline shrinkage, the gradient variance is significantly reduced.

## 5 CONCLUSION

In this paper, we propose a James-Stein-inspired baseline estimator for reinforcement learning with verifiable rewards (RLVR), which adaptively shrinks per-prompt reward estimates toward the global batch mean to reduce estimation variance. Our method is derived from a frequentist framework, requires no prior assumptions, and preserves unbiasedness via leave-one-out estimations. Theoretical derivations show that our estimator has lower expectations of MSE compared with non-shrinkage counterparts, provably leading to lower policy gradient variance. Empirical results show that our estimator consistently enhances RLVR training performance under different models, tasks and number of rollouts, with a significant reduction in terms of gradient variance. We hope the insight behind our approach inspires further improvements in critic-free RLVR algorithms.

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

## A    ADDITIONAL RELATED WORK

**Reinforcement Learning with Verifiable Rewards (RLVR).** RLVR refers to a training paradigm where the reward is computed by a rule-based verification function—typically indicating whether the model's final answer is correct. This approach has proven effective in enhancing the reasoning capabilities of LLMs. The recent success of RLVR is closely tied to advances in reinforcement learning algorithms, which can be broadly categorized into two groups. **Actor-critic methods**, such as PPO (Schulman et al., 2017) and its variants (e.g., VC-PPO (Yuan et al., 2025c), VinePPO (Kazemnejad et al., 2024), VAPO (Yuan et al., 2025b)), rely on training an additional value model (critic) to estimate baselines. While theoretically grounded, these methods incur high computational overhead. **Critic-free methods**, including RLOO (Ahmadian et al., 2024), ReMax (Li et al., 2023), GRPO (Shao et al., 2024), DAPO (Yu et al., 2025), and Dr.GRPO (Liu et al., 2025), eliminate the need for a learned value function by directly estimating baselines or advantages from multiple responses to the same prompt. These methods significantly reduce training cost and have become the dominant approach in practical RLVR pipelines. Their effectiveness largely hinges on the quality of the estimated baseline, which serves as a variance reduction tool in policy gradient updates.

**Baselines in Policy Gradient Methods.** The use of baselines in policy gradient methods was originally introduced (Williams, 1992) as a variance reduction technique without introducing bias. Early work formalized this as a control variate problem, showing that the optimal constant baseline is the average return (Weaver and Tao, 2013), while state-dependent baselines can further reduce variance (Greensmith et al., 2004). Actor-critic methods (Barto et al., 1989) extend this idea by learning value function approximations, and techniques such as generalized advantage estimation (GAE) (Schulman et al., 2015b) trade off bias and variance to improve stability and sample efficiency. More recent work explores state-action-dependent baselines: Q-Prop (Gu et al., 2016) leverages off-policy critics as control variates, and action-dependent factorized baselines (Wu et al., 2018) exploit policy structure to achieve lower variance in high-dimensional settings. Other methods use Stein's identity to learn expressive baselines with action dependency (Liu et al., 2017). However, later empirical study (Tucker et al., 2018) suggest that when value functions are well-approximated, simple state-dependent baselines can match or outperform more complex alternatives. In summary, an effective baseline should minimize variance, maintain zero or low bias, and be robust to implementation, with growing consensus emphasizing careful design over complexity.

**The James-Stein Estimator** The James-Stein estimator (James et al., 1961; Stein et al., 1956) is a classic estimator in frequentist statistics, showing that when simultaneously estimating the means of three or more independent Gaussian variables, the standard sample mean estimator is inadmissible under mean squared error (MSE). Specifically, the JS estimator improves estimation by shrinking each coordinate toward the global mean, thereby reducing the overall MSE. Brown (1971) discussed the admissibility of estimators for multivariate normal means, recurrent diffusions, and boundary value problems. Efron and Morris (1973) developed an empirical Bayes framework for Stein-type shrinkage rules and systematically compared Stein's estimator with a broad class of competitors. Feldman et al. (2014) further applied the James Stein estimator to multi-task learning. These line of surprising results have inspired a wide range of applications in empirical Bayes methods, shrinkage estimation, and high-dimensional statistics.

## B    FURTHER EXPERIMENTAL ANALYSIS

### B.1    REINFORCEMENT LEARNING FROM HUMAN FEEDBACKS

Beyond reasoning tasks, we further extend our JS baseline to another important LLM post-training setting: reinforcement learning from human feedback (RLHF). The goal of RLHF is to align an LLM's outputs with human preferences, which are usually modeled by a Bradley–Terry reward model. Following previous work (Hu, 2025; Mazoure et al., 2025; Zhang et al., 2025; Liu et al., 2024), we use a comprehensive training dataset (Dong et al., 2024) that contains 179k prompts from six widely used RLHF datasets: UltraFeedback (Cui et al., 2024), HelpSteer (Wang et al., 2024), OpenOrca (Lian et al., 2023), UltraInteract (Yuan et al., 2025a), DIBT-10K (Data is Better Together community, 2024), and Capybara Preferences (Argilla et al., 2024). We train a Llama-3.2-3B-Instruct model (Grattafiori et al., 2024) for one epoch using an 8B reward model (OpenRLHF, 2025) trained on 700k preference pairs, and evaluate on three widely used benchmarks: Arena-Hard-v0.1 (Li et al., 2024), Arena-Hard-v2.0 (Li et al., 2024), and Arena-Creative-Writing (Li et al., 2024). As shown in Table 2,

the JS baseline consistently outperforms the RLOO baseline on all three benchmarks. Thess results show that the effectiveness of JS baseline is not confined to rule-based, binary reward.

Table 2: RLHF benchmark results (win rates) of models after training on JS baseline, RLOO baseline and before training.

| Method | Arena-Hard v0.1 | Arena-Hard v2.0 | Arena-Creative-Writing |
|---|---|---|---|
| Llama-3.2-3B-It | 26.2% | 2.4% | 5.2% |
| RLOO | 55.3% | 5.6% | 20.2% |
| JS | 56.7% (+1.4%) | 7.4% (+1.8%) | 23.4% (+3.2%) |

## B.2 ABLATION STUDY ON DATASET DIFFICULTY HETEROGENEITY

We conduct an ablation study on the effect of the JS baseline under varying levels of dataset heterogeneity. For the Knights-and-Knaves dataset, a larger problem size (i.e., the number of propositions given) indicates higher complexity and difficulty. Therefore, we consider three training settings by mixing data with different difficulty levels: low heterogeneity (KnK-4 & KnK-5), medium heterogeneity (KnK-3 & KnK-7), and high heterogeneity (KnK-2 & KnK-9). For each dataset, we train a Qwen2.5-1.5B-Instruct model (Team, 2024) for 800 gradient steps and evaluate it on 200 questions drawn from the same distribution as the training data. Figure 9 shows the shrinkage coefficient for the different datasets, and Table 3 reports the test accuracy in the different settings. We observe that higher heterogeneity in the difficulty distribution leads to an adaptively smaller shrinkage, i.e., the algorithm behaves closer to vanilla RLOO. This adaptive adjustment mechanism yields an approximately optimal baseline across different training data distributions, resulting in consistent performance gains.

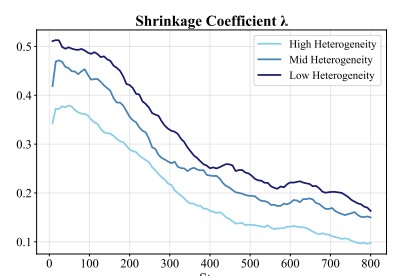

Figure 9: JS shrinkage coefficient during training in different difficulty heterogeneity.

| Method | Low | Mid | High |
|---|---|---|---|
| RLOO | 20.25 | 17.69 | 31.13 |
| JS | 25.69 | 23.85 | 34.25 |

Table 3: Average test set accuracy (%) under different heterogeneity settings.

## B.3 REDUCTION OF THE SQUARED L2 NORM WITH GROUND-TRUTH POLICY GRADIENT

We further obtain a precise estimate of the ground-truth policy gradient $\nabla_\theta J(\theta)$ and directly compare the variance of the policy gradient under different baselines using Equation (3). We first select 128 questions from the DAPO17k dataset and generate 256 responses for each question using the Qwen3-4B-Base model. For each question, by using all 256 rollouts to compute the policy gradient (Equation (1)), we obtain a precise estimate of the ground-truth $\nabla_\theta J(\theta)$. We then randomly sample 64 questions and $n$ (much smaller than 256) responses for each of them from the large response dataset we created. A small batch of $64 \times n$ samples is formed, similar to RL training setups in practice. Based on this small batch, we compute the policy gradients $g_{\text{RLOO}}$, $g_{\text{REINFORCE++}}$, $g_{\text{ReMax}}$ and $g_{\text{JS}}$ using different baselines, and compute the variance of the policy gradient by taking the squared L2 norm between $g$ and $\nabla_\theta J(\theta)$ (Equation (3)). In 10, we report the average squared L2 norm over 20 small batches with rollout numbers $n = 2, 4, 8$. We can see that the use of the JS baseline reduces the L2 norm between the estimated policy gradient and the ground truth by 12.5%, 8.6%, and 5.7%, respectively, providing direct evidence for improved training stability.

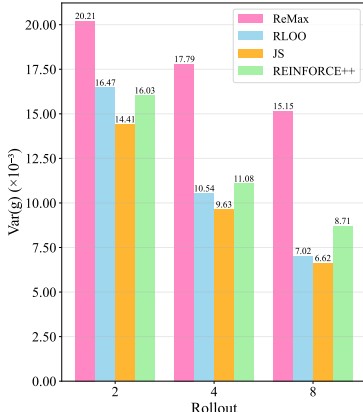

Figure 10: Comparison of different baselines on variance of policy gradient, obtained by computing squared L2 norm with ground truth policy gradients.

## B.4 ABLATION STUDY ON SEMANTIC HETEROGENEITY

In addition, we analyze how semantic heterogeneity of prompts affects variance reduction when using shrinkage baselines. We select three training datasets from our earlier experiments: KnK-4, Countdown-4, and a mixed dataset formed by combining 50% KnK-4 and 50% Countdown-4 problems. We run all experiments on Qwen2.5-7B-Instruct. By design, KnK-4 and Countdown-4 have similar pass rates and similar shrinkage coefficients on this model (Table 4). Therefore, mixing the two tasks **increases prompt semantic heterogeneity** while keeping the **reward distribution and shrinkage coefficient approximately matched**. Following the procedure in Appendix B.3, we estimate the variance of the policy-gradient estimator on Qwen2.5-7B-Instruct under different shrinkage coefficients $\lambda$ for all three datasets, with a fixed rollout number of 4. As shown in Figure 11, relative to the RLOO baseline ($\lambda = 0.0$), James–Stein (JS) shrinkage with $\lambda = 0.4$ reduces policy-gradient variance by 14.7% on KnK-4 and 11.8% on Countdown-4. On the mixed dataset, JS shrinkage still reduces variance, but the improvement is smaller (5.32%). However, $\lambda \approx 0.4$ remains the variance-minimizing choice in all three settings. These results suggest that semantic heterogeneity mainly influences the magnitude of variance reduction, while a James-Stein-style interpolation between the prompt-level mean and the batch-level mean continues to minimize gradient variance, regardless of how semantically heterogeneous the training data is.

| Dataset | Pass Rate (%) | $\lambda$ |
|---|---|---|
| KnK-4 | 13.35 | 0.43 |
| Countdown-4 | 14.62 | 0.41 |

Table 4: Qwen2.5-7B-Instruct pass rate (%) and shrinkage coefficient ($\lambda$) on two datasets.

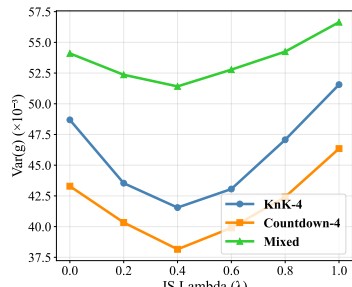

Figure 11: Effect of semantic heterogeneity on policy-gradient variance reduction with shrinkage baselines.

## C ADDITIONAL EXPERIMENTAL DETAILS

### C.1 JAMES-STEIN ADVANTAGE COMPUTATION

Below is the core python implementation of computing the advantage according to the James-Stein baseline. It only requires several lines of code and negligible additional computation.

Listing 1: James-Stein Advantage Estimator Python Implementation

```python
prompt_mean = torch.mean(rewards, dim=1)  # [n, m] => [n]
prompt_var = torch.var(rewards, dim=1, unbiased=True) / m
# Compute LOO batch mean and JS lambda
loo_means = [], js_lambdas = []
for i in range(n):
    other = torch.cat([prompt_mean[:i], prompt_mean[i + 1 :]])
    batch_loo_mean = torch.mean(other)
    v_square_i = torch.mean(torch.cat([prompt_var[:i], prompt_var[i + 1 :]]))
    s_square_i = torch.mean((other - batch_loo_mean) ** 2)
    js_lambda_i = v_square_i / (v_square_i + s_square_i)
    js_lambda_i *= (n - 1) / n
    loo_means.append(batch_loo_mean)
    js_lambdas.append(js_lambda_i)
loo_means = torch.stack(loo_means)  # [n]
js_lambdas = torch.stack(js_lambdas)  # [n]
rloo_baseline = (torch.sum(rewards, dim=1, keepdim=True) - rewards) / (m - 1)
    # [n, m]
js_baseline = rloo_baseline * (1 - js_lambdas[:, None]) + loo_means[:, None]
    * js_lambdas[:, None]
advantage = rewards - js_baseline
```

## C.2 DETAILS OF SECTION 4.1: MATH REASONING

Our dataset consists of 11,998 questions in MATH12k(Hendrycks et al., 2021) and 17,917 questions in DAPO17k(Yu et al., 2025). The detailed configurations of Qwen2.5-Math-1.5B, Qwen2.5-Math-7B and Qwen3-4B-Base are shown in Table 5 Table 12 and Table 13 respectively. An example of prompt template and its correct response are shown below.

---

**Qwen2.5-Math-7B**

`<|im_start|>`system
Please reason step by step and put the final answer in `\boxed{}`. `<|im_end|>`
`<|im_start|>`user Cities $A$ and $B$ are $45$ miles apart. Alicia lives in $A$ and Beth lives in $B$. Alicia bikes towards $B$ at 18 miles per hour. Leaving at the same time, Beth bikes toward $A$ at 12 miles per hour. How many miles from City $A$ will they be when they meet? Let's think step by step and output the final answer within `\boxed{}`. `<|im_end|>`
`<|im_start|>`assistant
To determine where Alicia and Beth will meet, we need to find out how long it will take for them to meet and then use that time to calculate the distance Alicia has traveled.

1. First, let's calculate their combined speed. Since Alicia bikes at 18 miles per hour and Beth bikes at 12 miles per hour, their combined speed is $18 + 12 = 30$ miles per hour.

2. Since the distance between the two cities is 45 miles, the time it takes for them to meet is the total distance divided by their combined speed. So, the time $t$ it takes for them to meet is

$$t = \frac{45}{30} = 1.5 \text{ hours}.$$

3. Now, we need to find out how far Alicia has traveled in 1.5 hours. Since Alicia bikes at 18 miles per hour, the distance she travels in 1.5 hours is
$$18 \times 1.5 = 27 \text{ miles}.$$

So, Alicia and Beth will meet 27 miles from City $A$. The final answer is

$$\boxed{27}$$

---

Table 5: Training setup of Qwen2.5-Math-1.5B

| Parameter | Value |
| --- | --- |
| Pretrained Model | Qwen2.5-Math-1.5B |
| Training Set | DAPO17k + MATH12k |
| Test Set | MATH500, AMC23, OlympiadBench |
| Prompts per batch | 64 |
| Generations per prompt | 4 |
| Gradient updates per RL step | 2 |
| Micro batch size | 2 |
| Max prompt length | 1024 |
| Max response length | 2048 |
| Lora Rank | 0 |
| Learning rate | $2 \times 10^{-6}$ |
| Clip ratio (high) | 0.22 |
| KL coefficient | 0.0 |
| Entropy coefficient | 0.0 |
| Rollout temperature | 0.8 |
| Validation temperature | 0.8 |
| Validation samples per prompt | 16 |
| Validation interval | 50 steps |
| Device | 4 NVIDIA GH200 |

C.3  DETAILS OF SECTION 4.2: LOGIC PUZZLE REASONING

We experiment on three representative logic puzzle tasks: Knights-and-Knaves, Countdown, and Maze. Knights-and-Knaves (KnK) is a classic logic puzzle where the goal is to determine truth-tellers (knights) and liars (knaves) based on their statements. Countdown is a numerical game where players use arithmetic operations on given numbers to reach a target value. Maze is a spatial puzzle that requires navigating through a complex grid of walls and paths to reach the finish point. For Countdown, we train on 10k questions with three numbers and 10k questions with four numbers, and evaluate on 200 questions from the same distribution. For Maze, we train on 20k 6×6 puzzles where the correct path length ranges from 4 to 8, and evaluate on 200 puzzles from the same distribution. For KnK, we train on two datasets: KnK-Easy, which consists of 10k problems each with 4, 5, or 6 people, and KnK-Hard, which consists of 10k problems each with 6, 7, or 8 people. For each experiment, we evaluate on 300 questions sampled from the same types of KnK puzzles used in training. We provide examples of question content and their corresponding correct responses for each puzzle below. Training configurations are summarized in Table 7 Table 8 Table 9 Table 10 and Table 14. For Knights-and-Knaves experiments, we also evaluated test accuracy of every subset (Table 6).

---

**Knights-and-Knaves**

**Question:**
A very special island is inhabited only by sages and fools. Sages always tell the truth, and fools always lie. You meet 6 inhabitants: Owen, Liam, Evelyn, William, Sofia, and Emily. As Owen put it, "William is a sage". Liam was heard saying, "William is a fool". Evelyn noted, "Sofia is a fool and Evelyn is a sage". "if Owen is a sage then Sofia is a sage," William mentioned. "if Liam is a sage then Evelyn is a sage," Sofia claimed. Emily told you that if Emily is a sage then Liam is a fool. So who is a sage and who is a fool? (Format your answer like: "Owen is a sage/fool, Liam is a sage/fool, Evelyn is a sage/fool, William is a sage/fool, Sofia is a sage/fool, and Emily is a sage/fool") Let's think step by step and output the final answer (should be a sentence) within \boxed{}.

**Response:**

- We need to consider two cases for Owen:
  - *Case 1: Owen is a sage.* If Owen is a sage, then his statement is true. Therefore, William must be a sage.
  - *Case 2: Owen is a fool.* If Owen is a fool, then his statement is false. Therefore, William must be a fool.
- **Liam's statement**: "William is a fool."
  - *Case 1: Liam is a sage.* If Liam is a sage, then his statement is true. Therefore, William must be a fool.
  - *Case 2: Liam is a fool.* If Liam is a fool, then his statement is false. Therefore, William must be a sage.
- **William's statement**: "If Owen is a sage then Sofia is a sage."
  Since Sofia is a sage, William's statement is true. Therefore, William must be a sage.
- **Owen's statement revisited**: "William is a sage."
  Since William is a sage, Owen's statement is true. Therefore, Owen must be a sage.
- **Liam's statement revisited**: "William is a fool."
  Since William is a sage, Liam's statement is false. Therefore, Liam must be a fool.
- **Emily's statement revisited**: "If Emily is a sage then Liam is a fool."
  Since Emily is a sage (from earlier), according to her statement, Liam must be a fool. This confirms our previous determination.
- **Owen's initial statement**: "William is a sage."
  Since Owen is a sage and William is a sage, Owen's statement is true. This is consistent.

**Therefore, summarizing:**

- Owen is a sage.
- Liam is a fool.
- Evelyn's statement was a contradiction, so Evelyn must be a fool.
- William is a sage.
- Sofia is a sage.
- Emily's statement was consistent, so Emily must be a sage.

$Owen$ is a sage, $Liam$ is a fool, $Evelyn$ is a fool, $William$ is a sage, $Sofia$ is a sage, and $Emily$ is a sage

---

### Countdown

**Question:**
Using the numbers [55, 53, 59, 22], create an equation that equals 71. You can use basic arithmetic operations (+, -, *, /) and each number can only be used once. Show your work in <think> </think> tags. And return the final answer in <answer> </answer> tags, for example <answer> (1 + 2) / 3 </answer>.

**Response:**
<think>

- We have the numbers 55, 53, 59, and 22. We need an equation that uses each at most once to make 71. Try combinations:

- Trials:

  1. $55 - 53 - 59 + 22 = -8 - 59 + 22 = -67 + 22 = -45$ (not it)
  2. $55 - 53 + 59 - 22 = 2 + 59 - 22 = 61 - 22 = 39$ (not it)
  3. $55 + 53 - 59 - 22 = 108 - 59 - 22 = 49 - 22 = 27$ (not it)
  4. $59 + 22 - 55 - 53 = 81 - 55 - 53 = 26 - 53 = -27$ (not it)
  5. $59 - 22 + 53 - 55 = 37 + 53 - 55 = 90 - 55 = 35$ (not it)
  6. $53 + 22 + 55 - 59 = 75 + 55 - 59 = 130 - 59 = \boxed{71}$ (works)

- Therefore, an expression that equals 71 is $53 + 22 + 55 - 59$.

</think>
<answer> 53+22+55-59 </answer>

### Maze

**Question:**
You need to solve the following maze. '*' denotes the wall that you cannot walk through, '.' denotes available area that you can walk through. 'S' denotes the starting point, 'E' denotes the destination. There are four movement actions, including Left, Right, Up, Down. You need to use L to denote Left movement, R to denote Right movement, U to denote Up movement, and D to denote Down movement. You can analyze the maze to find the correct path, and you should write the final path in the <answer> </answer>, e.g., <answer> LLRRDUL </answer>.
## Maze
```
******
*.*.E*
*....*
**.*.*
**.S.*
******
```
Now try to analyze the maze and put the final path in the <answer> </answer>.

**Response:**

1. **Locate the start and end.** Index rows from top to bottom and columns from left to right, starting at 1.

   - Start $S$ at $(5, 4)$ (row 5: `**.S.*`, column 4 is `S`).
   - End $E$ at $(2, 5)$ (row 2: `*.*.E*`, column 5 is `E`).

2. **Check immediate moves from** $S(5, 4)$**.**

   - $U : (4, 4)$ — row 4 col 4 is `*`, blocked.
   - $D : (6, 4)$ — row 6 is `******`, blocked.
   - $L : (5, 3)$ — row 5 col 3 is `.`, open.
   - $R : (5, 5)$ — row 5 col 5 is `.`, open.

3. **Choose a straightforward open route to approach** $E$**.**

   - Move $R$: $(5, 4) \rightarrow (5, 5)$, cell `.` (open).
   - Move $U$: $(5, 5) \rightarrow (4, 5)$, cell `.` (open).
   - Move $U$: $(4, 5) \rightarrow (3, 5)$, cell `.` (open).
   - Move $U$: $(3, 5) \rightarrow (2, 5)$, cell `E` (destination reached).

4. **Validate the sequence.**

   - Each step lands on an open cell (`.`) until the final step reaches `E`.
   - No step crosses a wall (`*`).
   - The sequence uses only the allowed moves $\{R, U, U, U\}$.

   <answer> RUUU </answer>

Table 6: Detailed test accuracy (%) on different subsets for Knights-and-Knaves experiments. Larger number in KnK means the task is more complex.

| Algorithm | Train on KnK-Easy | | | Train on KnK-Hard | | |
|---|---|---|---|---|---|---|
| | KnK-4 | KnK-5 | KnK-6 | KnK-6 | KnK-7 | KnK-8 |
| RLOO | 58.38 | 49.89 | 41.25 | 39.50 | 30.00 | 16.75 |
| JS | 75.87 | 66.16 | 55.25 | 42.13 | 32.50 | 25.38 |

Table 7: Training setup for KnK-Easy

| Parameter | Value |
|---|---|
| Pretrained Model | Qwen2.5-7B-Instruct |
| Training Set | KnK-4, KnK-5, KnK-6 |
| Test Set | KnK-4-Test, KnK-5-Test, KnK-6-Test |
| Prompts per batch | 32 |
| Generations per prompt | 8 |
| Gradient updates per RL step | 2 |
| Micro batch size | 2 |
| Max prompt length | 1024 |
| Max response length | 2048 |
| Lora Rank | 256 |
| Learning rate | $4 \times 10^{-5}$ |
| Clip ratio (high) | 0.22 |
| KL coefficient | 0.0 |
| Entropy coefficient | 0.0 |
| Rollout temperature | 0.7 |
| Validation temperature | 0.7 |
| Validation samples per prompt | 16 |
| Validation interval | 25 steps |
| Device | 4 NVIDIA GH200 |

Table 8: Training setup for KnK-Hard

| Parameter | Value |
|---|---|
| Pretrained Model | Qwen2.5-7B-Instruct |
| Training Set | KnK-6, KnK-7, KnK-8 |
| Test Set | KnK-6-Test, KnK-7-Test, KnK-8-Test |
| Prompts per batch | 32 |
| Generations per prompt | 8 |
| Gradient updates per RL step | 2 |
| Micro batch size | 2 |
| Max prompt length | 1024 |
| Max response length | 2048 |
| Lora Rank | 256 |
| Learning rate | $4 \times 10^{-5}$ |
| Clip ratio (high) | 0.22 |
| KL coefficient | 0.0 |
| Entropy coefficient | 0.0 |
| Rollout temperature | 0.7 |
| Validation temperature | 0.7 |
| Validation samples per prompt | 16 |
| Validation interval | 25 steps |
| Device | 4 NVIDIA GH200 |

Table 9: Training setup for Countdown

| Parameter | Value |
| --- | --- |
| Pretrained Model | Qwen2.5-3B |
| Training Set | Countdown3, Countdown4 |
| Test Set | Countdown3-Test, Countdown4-Test |
| Prompts per batch | 64 |
| Generations per prompt | 5 |
| Gradient updates per RL step | 2 |
| Micro batch size | 4 |
| Max prompt length | 512 |
| Max response length | 1024 |
| Lora Rank | 0 |
| Learning rate | $1 \times 10^{-6}$ |
| Clip ratio (high) | 0.22 |
| KL coefficient | 0.0 |
| Entropy coefficient | 0.0 |
| Rollout temperature | 0.7 |
| Validation temperature | 0.7 |
| Validation samples per prompt | 16 |
| Validation interval | 10 steps |
| Device | 2 NVIDIA GH200 |

Table 10: Training setup for Maze

| Parameter | Value |
| --- | --- |
| Pretrained Model | Ministral-8B-Instruct |
| Training Set | Maze6x6 |
| Test Set | Maze6x6-Test |
| Prompts per batch | 32 |
| Generations per prompt | 8 |
| Gradient updates per RL step | 2 |
| Micro batch size | 8 |
| Max prompt length | 1024 |
| Max response length | 2048 |
| Lora Rank | 0 |
| Learning rate | $3 \times 10^{-7}$ |
| Clip ratio (high) | 0.25 |
| KL coefficient | 0.0 |
| Entropy coefficient | 0.0 |
| Rollout temperature | 0.7 |
| Validation temperature | 0.7 |
| Validation samples per prompt | 16 |
| Validation interval | 25 steps |
| Device | 4 NVIDIA GH200 |

Table 11: Training Setup for GSM8k

| Parameter | Value |
|---|---|
| Pretrained Model | Qwen2.5-0.5B-Instruct |
| Training Set | GSM8k-Train |
| Test Set | GSM8k-Test |
| Prompts per batch | 64 |
| Generations per prompt | 4 |
| Gradient updates per RL step | 1 |
| Max prompt length | 1024 |
| Max response length | 2048 |
| Learning rate | $1 \times 10^{-6}$ |
| Clip ratio | 0.2 |
| KL coefficient | 0.0 |
| Entropy coefficient | 0.0 |
| Rollout temperature | 0.7 |
| Validation temperature | 0.5 |
| Validation samples per prompt | 10 |
| Validation interval | 100 steps |
| Device | 1 NVIDIA GH200 |

Table 12: Training setup of Qwen2.5-Math-7B

| Parameter | Value |
|---|---|
| Pretrained Model | Qwen2.5-Math-7B |
| Training Set | DAPO17k + MATH12k |
| Test Set | MATH500, AMC23, OlympiadBench |
| Prompts per batch | 64 |
| Generations per prompt | 4 |
| Gradient updates per RL step | 2 |
| Micro batch size | 2 |
| Max prompt length | 1024 |
| Max response length | 2048 |
| Lora Rank | 256 |
| Learning rate | $2 \times 10^{-5}$ |
| Clip ratio (high) | 0.22 |
| KL coefficient | 0.0 |
| Entropy coefficient | 0.0 |
| Rollout temperature | 0.8 |
| Validation temperature | 0.8 |
| Validation samples per prompt | 16 |
| Validation interval | 50 steps |
| Device | 4 NVIDIA GH200 |

Table 13: Training setup of Qwen3-4B-Base

| Parameter | Value |
| --- | --- |
| Pretrained Model | Qwen3-4B-Base |
| Training Set | DAPO17k |
| Test Set | MATH500, AMC23, OlympiadBench |
| Prompts per batch | 64 |
| Generations per prompt | 4 |
| Gradient updates per RL step | 2 |
| Micro batch size | 2 |
| Max prompt length | 1024 |
| Max response length | 3072 |
| Lora Rank | 0 |
| Learning rate | $2 \times 10^{-6}$ |
| Clip ratio (high) | 0.28 |
| KL coefficient | 0.0 |
| Entropy coefficient | 0.0 |
| Rollout temperature | 1.0 |
| Validation temperature | 1.0 |
| Validation samples per prompt | 16 |
| Validation interval | 50 steps |
| Device | 4 NVIDIA H100 |

Table 14: Training setup of Qwen2.5-1.5B-Instruct

| Parameter | Value |
| --- | --- |
| Pretrained Model | Qwen2.5-1.5B-Instruct |
| Training Set | KnK-2, KnK-3 |
| Test Set | KnK-2-Test, KnK-3-Test |
| Prompts per batch | 32 |
| Generations per prompt | 8 |
| Gradient updates per RL step | 2 |
| Micro batch size | 2 |
| Max prompt length | 1024 |
| Max response length | 2048 |
| Lora Rank | 0 |
| Learning rate | $5 \times 10^{-7}$ |
| Clip ratio (high) | 0.22 |
| KL coefficient | 0.0 |
| Entropy coefficient | 0.0 |
| Rollout temperature | 0.7 |
| Validation temperature | 0.7 |
| Validation samples per prompt | 16 |
| Validation interval | 25 steps |
| Device | 4 NVIDIA H100 |

## C.4  DETAILS OF SECTION 4.3: COMPARISON WITH DIFFERENT BASELINES

We train Qwen2.5-0.5B-Instruct model on GSM8k dataset for 500 steps. GSM8k consists of 7,473 questions in the training set and 1,319 in test set. For each rollout batch, we sample 64 distinct prompts, and for each prompt we generate $m \in \{2, 4, 8\}$ responses with official template of GSM8k. Each experiment is repeated across five random seeds $\{0,1,2,3,4\}$. Summary of hyperparameters and configurations is provided in Table 11. Detailed numbers are in Table 15, and the detailed curves of validation accuracy are in Figure 12.

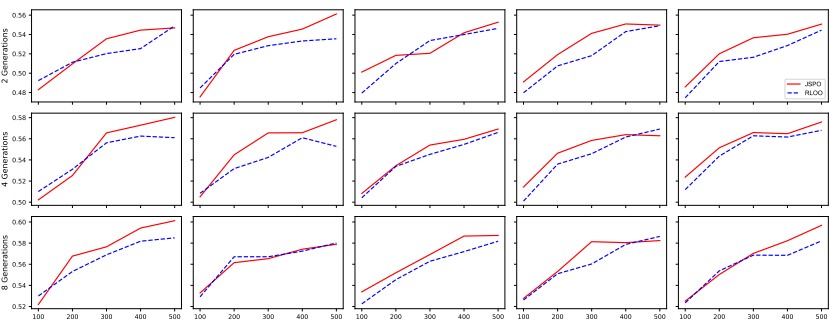

Figure 12: All the single experiments between JS Baseline and RLOO Baseline.

Table 15: Test Accuracy (%) across 5 runs for different algorithms under 2, 4, and 8 Generations. Each cell shows 5 accuracy values (%) at steps 100 to 500. Initial test accuracy is 40.03% for all.

| Baseline | 2 Generations | 4 Generations | 8 Generations |
|---|---|---|---|
| JS | 48.29, 50.96, 53.56, 54.45, 54.68
47.55, 52.36, 53.77, 54.56, 56.12
50.11, 51.84, 52.06, 54.18, 55.27
49.10, 51.92, 54.12, 55.09, 54.97
48.58, 52.01, 53.66, 54.03, 55.07 | 50.22, 52.53, 56.56, 57.29, 58.03
50.51, 54.48, 56.56, 56.57, 57.80
50.83, 53.45, 55.41, 55.95, 56.92
51.43, 54.63, 55.85, 56.39, 56.29
52.37, 55.15, 56.59, 56.48, 57.59 | 52.19, 56.77, 57.65, 59.42, 60.12
53.28, 56.15, 56.53, 57.42, 57.89
53.39, 55.19, 56.92, 58.67, 58.73
52.77, 55.31, 58.13, 58.04, 58.24
52.51, 55.03, 57.03, 58.23, 59.68 |
| BLOO | 49.42, 50.80, 51.88, 52.23, 54.61
48.88, 51.43, 52.32, 53.39, 53.39
48.22, 50.42, 52.99, 54.41, 54.30
48.08, 49.95, 53.15, 53.43, 53.06
48.57, 51.46, 52.08, 53.98, 55.62 | 49.32, 52.65, 54.19, 55.16, 56.13
50.46, 53.39, 55.85, 55.11, 55.90
49.64, 52.43, 54.36, 54.47, 55.95
50.28, 52.86, 54.53, 54.51, 55.98
49.55, 52.53, 54.71, 54.91, 56.35 | 52.10, 54.61, 55.91, 56.91, 57.54
51.68, 54.59, 56.63, 56.07, 57.32
50.99, 54.25, 55.13, 55.68, 57.02
51.33, 54.71, 56.15, 55.16, 57.47
51.08, 54.98, 56.06, 56.95, 56.74 |
| RLOO | 49.23, 51.13, 52.02, 52.54, 54.89
48.48, 51.95, 52.84, 53.33, 53.56
47.95, 50.99, 53.37, 53.99, 54.63
47.98, 50.75, 51.81, 54.30, 54.91
47.46, 51.21, 51.64, 52.86, 54.45 | 51.01, 53.12, 55.62, 56.26, 56.10
50.84, 53.18, 54.24, 56.10, 55.28
50.42, 53.39, 54.53, 55.47, 56.60
50.10, 53.60, 54.59, 56.16, 56.92
51.19, 54.36, 56.29, 56.16, 56.80 | 52.99, 55.31, 56.89, 58.18, 58.50
52.92, 56.71, 56.71, 57.24, 58.01
52.25, 54.54, 56.29, 57.20, 58.18
52.62, 55.10, 56.03, 57.88, 58.63
52.34, 55.38, 56.86, 56.85, 58.21 |
| GRPO | 48.67, 51.99, 52.99, 54.72, 54.41
47.73, 52.42, 53.15, 54.06, 52.87
48.92, 51.08, 54.24, 54.12, 54.34
48.57, 51.72, 50.70, 52.89, 53.01
48.77, 51.52, 53.24, 53.49, 53.79 | 51.39, 53.54, 55.56, 55.94, 55.54
51.31, 52.96, 55.15, 55.79, 57.39
49.75, 53.51, 54.38, 54.57, 55.21
50.05, 54.43, 56.62, 57.77, 56.88
51.63, 53.04, 54.47, 55.09, 56.19 | 51.96, 56.07, 57.10, 58.27, 58.20
52.52, 55.53, 56.51, 57.80, 59.06
52.43, 55.57, 57.12, 58.37, 59.05
52.43, 55.77, 57.56, 57.54, 57.21
52.48, 55.31, 57.83, 57.85, 57.91 |
| ReMax | 49.16, 52.12, 52.40, 52.54, 54.83
48.76, 52.68, 53.37, 54.39, 54.10
48.79, 52.10, 52.45, 53.75, 55.65
48.57, 51.52, 52.98, 52.40, 54.06
48.92, 51.63, 52.54, 53.33, 54.86 | 50.05, 53.37, 55.35, 55.95, 56.33
49.08, 51.15, 53.92, 55.69, 55.94
50.19, 52.83, 54.98, 54.59, 57.12
48.78, 51.31, 54.12, 54.98, 56.57
49.48, 53.80, 54.16, 55.25, 56.40 | 51.04, 54.71, 56.65, 57.10, 57.81
52.01, 55.71, 56.29, 57.10, 59.38
51.55, 54.22, 55.66, 57.26, 57.62
51.05, 54.39, 55.69, 55.36, 56.19
51.32, 54.84, 55.99, 56.87, 57.81 |
| REINFORCE++ | 48.86, 51.90, 53.80, 53.57, 55.01
49.61, 52.18, 53.06, 54.86, 55.27
48.19, 50.54, 52.66, 53.74, 53.51
49.51, 51.96, 52.27, 54.63, 55.33
48.76, 51.35, 52.46, 53.98, 54.96 | 50.05, 52.87, 53.77, 55.44, 56.79
49.95, 53.25, 54.98, 54.97, 53.69
49.11, 53.33, 55.19, 55.72, 56.04
49.63, 52.24, 53.72, 55.13, 54.01
49.52, 52.68, 54.86, 55.21, 55.82 | 51.51, 54.50, 55.59, 58.10, 57.92
51.17, 54.57, 57.22, 57.15, 57.24
52.08, 55.82, 57.16, 57.13, 57.00
51.95, 54.84, 57.03, 57.77, 57.04
50.99, 53.82, 56.94, 57.23, 57.32 |

## C.5 DETAILS OF SECTION 4.5: TRAINING DYNAMICS

### C.5.1 PROOF OF EQUATION (15)

Let i.i.d. micro-batch gradients $g_i \in \mathbb{R}^P$ for $i = 1, \ldots, m$, with $\mu = \mathbb{E}[g_i]$ and $\Sigma = \mathrm{Cov}(g_i)$. Let $g = \frac{1}{m} \sum_{i=1}^{m} g_i$ denote the batch gradient, and $\bar{g} = \frac{1}{m} \sum_{i=1}^{m} g_i$ the sample mean.

We first express the target quantity:

$$\mathrm{Var}(g) = \mathrm{Tr}\big(\mathrm{Cov}(g)\big) = \mathrm{Tr}\Big(\mathrm{Cov}\big(\tfrac{1}{m} \sum_{i=1}^{m} g_i\big)\Big) = \mathrm{Tr}\Big(\tfrac{1}{m^2} \sum_{i=1}^{m} \mathrm{Cov}(g_i)\Big) = \frac{1}{m} \mathrm{Tr}(\Sigma).$$

Using the identity $\sum_{i=1}^{m} \|g_i - \bar{g}\|^2 = \sum_{i=1}^{m} \|g_i\|^2 - m\|\bar{g}\|^2$, we take expectations term by term:

$$\mathbb{E}\left[\sum_{i=1}^{m} \|g_i - \bar{g}\|^2\right] = \sum_{i=1}^{m} \mathbb{E}\|g_i\|^2 - m\,\mathbb{E}\|\bar{g}\|^2.$$

For the two moments, we have

$$\mathbb{E}\|g_i\|^2 = \mathrm{Tr}(\Sigma) + \|\mu\|^2, \qquad \mathbb{E}\|\bar{g}\|^2 = \mathbb{E}\Big\|\frac{1}{m} \sum_{i=1}^{m} g_i\Big\|^2 = \frac{1}{m} \mathrm{Tr}(\Sigma) + \|\mu\|^2.$$

Substituting back gives

$$\mathbb{E}\left[\sum_{i=1}^{m} \|g_i - \bar{g}\|^2\right] = m\big(\mathrm{Tr}(\Sigma) + \|\mu\|^2\big) - m\Big(\frac{1}{m} \mathrm{Tr}(\Sigma) + \|\mu\|^2\Big) = (m-1)\mathrm{Tr}(\Sigma).$$

Dividing by $(m-1)$ yields the unbiased sample-trace of $\Sigma$:

$$\mathbb{E}\left[\frac{1}{m-1} \sum_{i=1}^{m} \|g_i - \bar{g}\|^2\right] = \mathrm{Tr}(\Sigma).$$

Since $\mathrm{Var}(g) = \frac{1}{m} \mathrm{Tr}(\Sigma)$, multiplying by $\frac{1}{m}$ produces the desired unbiased estimator of $\mathrm{Var}(g)$:

$$\mathbb{E}\left[\frac{1}{m} \cdot \frac{1}{m-1} \sum_{i=1}^{m} \|g_i - \bar{g}\|^2\right] = \frac{1}{m} \mathrm{Tr}(\Sigma) = \mathrm{Var}(g).$$

Finally, using $\sum_{i=1}^{m} \|g_i - \bar{g}\|^2 = \sum_{i=1}^{m} \|g_i\|^2 - \frac{1}{m}\big\|\sum_{i=1}^{m} g_i\big\|^2$, the estimator can be written in the form of:

$$\widehat{\mathrm{Var}(g)} = \frac{1}{m} \cdot \frac{1}{m-1}\left(\sum_{i=1}^{m} \|g_i\|^2 - \frac{1}{m}\Big\|\sum_{i=1}^{m} g_i\Big\|^2\right).$$

### C.5.2 PSEUDO CODE OF GRADIENT VARIANCE ESTIMATION

```
sum_sq = 0.0                   # accumulates sum_i ||g_i||^2
sum_g = zeros_like(vector)     # accumulates sum_i g_i  (shape: (P,))
for i in range(1, m+1):
    g_i = compute_flattened_gradient_for_microbatch(i)  # shape: (P,)
    sum_sq += dot(g_i, g_i)       # scalar: ||g_i||^2
    sum_g  += g_i                 # vector: sum of g_i

# sample-trace of covariance at micro-batch level: (1/(m-1)) * sum_i ||g_i -
    g_bar||^2
trace_cov_micro = ( sum_sq - dot(sum_g, sum_g) / m ) / (m - 1)

# unbiased estimate of batch-gradient variance trace Var(g) = tr(Cov(g))
# because Cov(g) = (1/m) * Cov(g_i)
var_g_trace_estimate = (1.0 / m) * trace_cov_micro
```

## C.6 ADDITIONAL EXPERIMENTAL RESULTS OF SECTION 4

We aggregate more experimental details in this section, including training reward curves (Figure 13), additional shrinkage coefficient curves (Figure 14), and additional variance estimation curves (Figure 15).

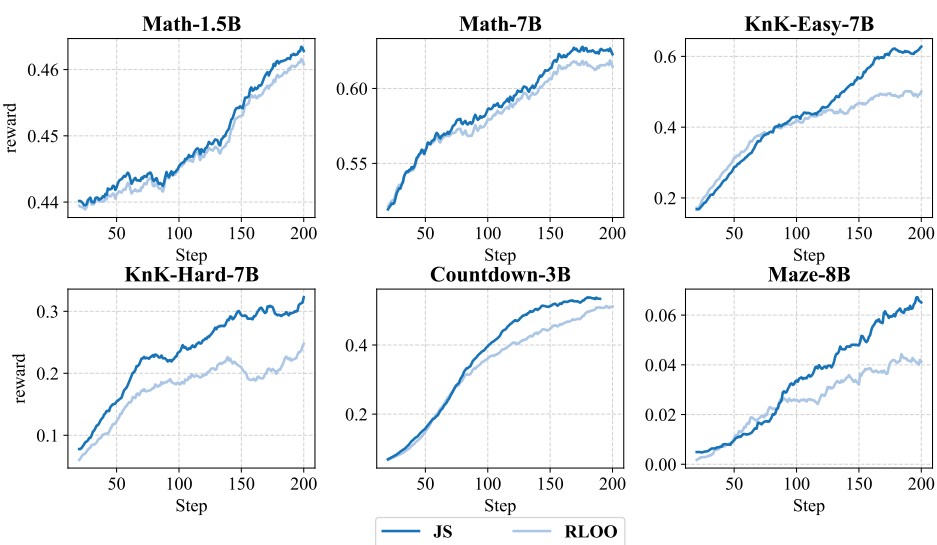

Figure 13: Moving average reward curves during training, compared with JS baseline and RLOO baseline.

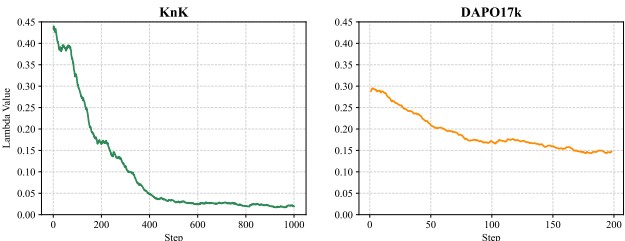

Figure 14: The moving average shrinkage coefficient curves on different datasets. Left: KnK dataset; Right: DAPO17k dataset. Adaptive shrinkage enables dynamically optimal baseline estimation that adjusts to the data distributions and training progresses.

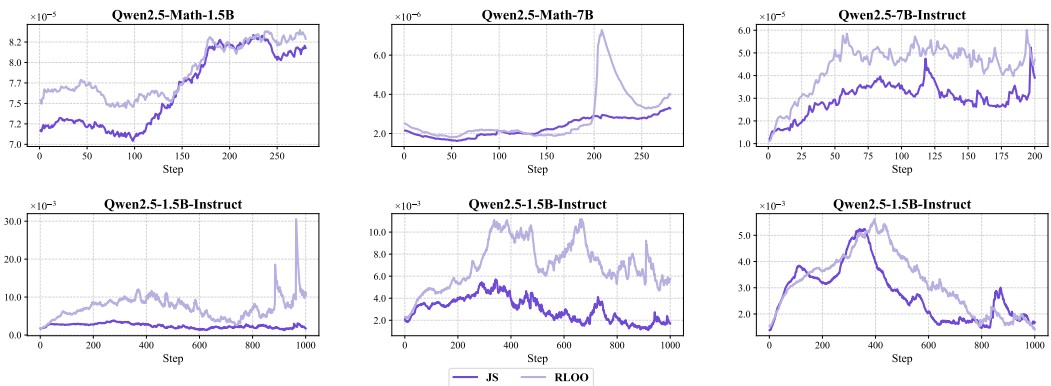

Figure 15: Additional running average gradient variance curves during training.

## D    PROOFS OF THEORETICAL RESULTS

In this section, we use the notation $\mathbf{Y} - y_i^j$ to denote the set of all $y_{i'}^{j'}$ in which $(i', j') \neq (i, j)$.

### D.1    PROOF OF PROPOSITION 1

Consider gradient update on each sample $\mathbb{E}_{\mathbf{x}, \mathbf{Y}}[(r_i^j - b_i^j)\nabla_\theta \log \pi_\theta(y_i^j | x_i)]$. We have

$$\mathbb{E}_{\mathbf{x}, \mathbf{Y}}[(r_i^j - b_i^j)\nabla_\theta \log \pi_\theta(y_i^j | x_i)]$$

$$= \mathbb{E}_{\mathbf{x}, \mathbf{Y}}[r_i^j \nabla_\theta \log \pi_\theta(y_i^j | x_i)] - \mathbb{E}_{\mathbf{x}, \mathbf{Y}}[b_i^j \nabla_\theta \log \pi_\theta(y_i^j | x_i)]$$

$$= \nabla_\theta J(\theta) - \mathbb{E}_{\mathbf{x}, \mathbf{Y} - y_i^j} \mathbb{E}_{y_i^j \sim \pi_\theta(\cdot | x_i)}[b_i^j \nabla_\theta \log \pi_\theta(y_i^j | x_i)]$$

$$= \nabla_\theta J(\theta) - \mathbb{E}_{\mathbf{x}, \mathbf{Y} - y_i^j} \left\{ b_i^j \mathbb{E}_{y_i^j \sim \pi_\theta(\cdot | x_i)}[\nabla_\theta \log \pi_\theta(y_i^j | x_i)] \right\}$$

$$= \nabla_\theta J(\theta) - \mathbb{E}_{\mathbf{x}, \mathbf{Y} - y_i^j} \left\{ b_i^j \sum_{y_i^j}[\nabla_\theta \pi_\theta(y_i^j | x_i)] \right\}$$

$$= \nabla_\theta J(\theta).$$

Since this holds for all $i, j$, we have

$$\mathbb{E}_{\mathbf{x}, \mathbf{Y}}[g(\mathbf{x}, \mathbf{Y}; \theta)] = \frac{1}{n} \sum_{i=1}^n \frac{1}{m} \sum_{j=1}^m \mathbb{E}_{\mathbf{x}, \mathbf{Y}}[(r_i^j - b_i^j)\nabla_\theta \log \pi_\theta(y_i^j | x_i)]$$

$$= \nabla_\theta J(\theta).$$

### D.2    PROOF OF PROPOSITION 2

Note that $b_i^{j,\text{JS1}} = b_i^{1,\text{JS1}}$ for all $i, j$. We can also rewrite the interpolation as

$$b_i^{1,\text{JS1}} = \left(1 - \frac{n-1}{n}\lambda\right) \widehat{\mu}_i + \frac{n-1}{n}\lambda \widehat{\overline{\mu}}_{-i}.$$

We can let $\gamma = \frac{n-1}{n}\lambda$. So we can rewrite the objective as

$$\text{MSE}^{\text{relax}} = \mathbb{E}_{\mathbf{Y}}\left[\frac{1}{n}\sum_{i=1}^n \left(\mu_i - b_i^{1,\text{JS1}}\right)^2\right]$$

$$= \frac{1}{n}\sum_{i=1}^n \mathbb{E}_{\mathbf{Y}}\left[((1-\gamma)(\mu_i - \widehat{\mu}_i) + \gamma(\mu_i - \widehat{\overline{\mu}}_{-i}))^2\right]$$

$$= \frac{1}{n}\sum_{i=1}^n \left\{(1-\gamma)^2 \mathbb{E}_{\mathbf{Y}}\left[(\mu_i - \widehat{\mu}_i)^2\right] + 2\gamma(1-\gamma)\mathbb{E}_{\mathbf{Y}}\left[(\mu_i - \widehat{\mu}_i)(\mu_i - \widehat{\overline{\mu}}_{-i})\right] + \gamma^2 \mathbb{E}_{\mathbf{Y}}\left[(\mu_i - \widehat{\overline{\mu}}_{-i})^2\right]\right\}$$

$$= \frac{1}{n}\sum_{i=1}^n \left\{(1-\gamma)^2 \text{Var}_{\mathbf{Y}}[\widehat{\mu}_i] + 2\gamma(1-\gamma)\mathbb{E}_{\mathbf{Y}}\left[\mu_i - \widehat{\mu}_i\right]\mathbb{E}_{\mathbf{Y}}\left[\mu_i - \widehat{\overline{\mu}}_{-i}\right] + \gamma^2 \mathbb{E}_{\mathbf{Y}}\left[(\mu_i - \widehat{\overline{\mu}}_{-i})^2\right]\right\}$$

$$= \frac{1}{n}\sum_{i=1}^n \left\{(1-\gamma)^2 \frac{1}{m}\sigma_i^2 + \gamma^2 \mathbb{E}_{\mathbf{Y}}\left[(\mu_i - \widehat{\overline{\mu}}_{-i})^2\right]\right\}.$$

Let $\bar{\mu}_{-i} := \frac{1}{n-1}\sum_{i' \neq i} \mu_{i'}$. For the second term in the summation, note that

$$\mathbb{E}_{\mathbf{Y}}\left[(\mu_i - \widehat{\overline{\mu}}_{-i})^2\right] = \mathbb{E}_{\mathbf{Y}}\left[((\mu_i - \bar{\mu}_{-i}) + (\bar{\mu}_{-i} - \widehat{\overline{\mu}}_{-i}))^2\right]$$

$$= (\mu_i - \bar{\mu}_{-i})^2 + \mathbb{E}_{\mathbf{Y}}\left[(\bar{\mu}_{-i} - \widehat{\overline{\mu}}_{-i})^2\right]$$

$$= \left(\frac{n}{n-1}\right)^2 (\mu_i - \bar{\mu})^2 + \text{Var}_{\mathbf{Y}}\left[\widehat{\overline{\mu}}_{-i}\right]$$

$$= \left(\frac{n}{n-1}\right)^2 (\mu_i - \bar{r})^2 + \frac{1}{(n-1)^2}\sum_{i' \neq i} \text{Var}_{\mathbf{Y}}[\widehat{\mu}_i].$$

Therefore, we have

$$
\mathbb{E}\left[\frac{1}{n}\sum_{i=1}^{n}\left(\mu_i - b_i^{1,\mathrm{JS1}}\right)^2\right]
$$

$$
=\frac{1}{n}\sum_{i=1}^{n}\left\{(1-\gamma)^2\frac{1}{m}\sigma_i^2 + \gamma^2\left[\left(\frac{n}{n-1}\right)^2(\mu_i - \bar{\mu})^2 + \frac{1}{(n-1)^2}\sum_{i'\neq i}\frac{1}{m}\sigma_i^2\right]\right\}
$$

$$
=\frac{1}{n}(1-\gamma)^2\sum_{i=1}^{n}\frac{1}{m}\sigma_i^2 + \frac{n}{(n-1)^2}\gamma^2\sum_{i=1}^{n}(\mu_i - \bar{\mu})^2 + \frac{1}{n(n-1)^2}\gamma^2\sum_{i=1}^{n}\sum_{i'\neq i}\frac{1}{m}\sigma_i^2
$$

$$
=\frac{1}{n}(1-\gamma)^2\sum_{i=1}^{n}\frac{1}{m}\sigma_i^2 + \frac{n}{(n-1)^2}\gamma^2\sum_{i=1}^{n}(\mu_i - \bar{\mu})^2 + \frac{1}{n(n-1)}\gamma^2\sum_{i=1}^{n}\frac{1}{m}\sigma_i^2
$$

$$
=\frac{n}{n-1}(s+v)\gamma^2 - 2v\gamma + v.
$$

This is a quadratic function of $\gamma$, and so the global minimum is $\gamma^* := \frac{n-1}{n}\frac{v}{s+v}$.

### D.3 PROOF OF THEOREM 1

Recall Equation (10):

$$
b_i^{j,\mathrm{JS2}} = (1-\lambda_i^j)\widehat{\mu}_i^{-j} + \lambda_i^j\widehat{\bar{\mu}}_{-i}.
$$

Since each baseline $b_i^{j,\mathrm{JS2}}$ has its own James-Stein coefficient $\lambda_i^j$, we only need to minimize each square error term $\mathbb{E}_{\mathbf{x},\mathbf{Y}}[\mu_i - b_i^{j,\mathrm{JS2}}]$ in order to minimize the whole objective MSE.

Then for any $i, j$, we have

$$
\mathbb{E}_{\mathbf{x},\mathbf{Y}}[(\mu_i - b_i^j)^2]
$$
$$
=\mathbb{E}_{\mathbf{x},\mathbf{Y}}[((1-\lambda_i^j)(\mu_i - \widehat{\mu}_i^{-j}) + \lambda_i^j(\mu_i - \widehat{\bar{\mu}}_{-i}))^2]
$$
$$
=\mathbb{E}_{\mathbf{x}}\mathbb{E}_{\mathbf{Y}}\left[(1-\lambda_i^j)^2(\mu_i - \widehat{\mu}_i^{-j})^2 + (\lambda_i^j)^2(\mu_i - \widehat{\bar{\mu}}_{-i})^2 + 2\lambda_i^j(1-\lambda_i^j)(\mu_i - \widehat{\mu}_i^{-j})(\mu_i - \widehat{\bar{\mu}}_{-i})\right]
$$
$$
=\mathbb{E}_{\mathbf{X}}\left\{(1-\lambda_i^j)^2\mathbb{E}_{\mathbf{Y}}\left[(\mu_i - \widehat{\mu}_i^{-j})^2\right] + (\lambda_i^j)^2\mathbb{E}_{\mathbf{Y}}\left[(\mu_i - \widehat{\bar{\mu}}_{-i})^2\right] + 2\lambda_i^j(1-\lambda_i^j)\mathbb{E}_{\mathbf{Y}}\left[\mu_i - \widehat{\mu}_i^{-j}\right]\mathbb{E}_{\mathbf{Y}}\left[\mu_i - \widehat{\bar{\mu}}_{-i}\right]\right\}
$$
$$
=\mathbb{E}_{\mathbf{X}}\left\{(1-\lambda_i^j)^2\mathbb{E}_{\mathbf{Y}}\left[(\mu_i - \widehat{\mu}_i^{-j})^2\right] + (\lambda_i^j)^2\mathbb{E}_{\mathbf{Y}}\left[(\mu_i - \widehat{\bar{\mu}}_{-i})^2\right]\right\}.
$$

For the first term in the summation, we have

$$
\mathbb{E}_{\mathbf{Y}}\left[(\mu_i - \widehat{\mu}_i^{-j})^2\right] = \mathrm{Var}_{\mathbf{Y}}[\widehat{\mu}_i^{-j}] = \frac{1}{m-1}\sigma^2(x_i).
$$

Recall that $\sigma^2(x_i) = \mathrm{Var}_{y\sim\pi_\theta(\cdot|x_i)}[r(x_i,y)]$. For the second term, we have

$$
\mathbb{E}_{\mathbf{Y}}\left[(\mu_i - \widehat{\bar{\mu}}_{-i})^2\right] = \mathbb{E}_{\mathbf{Y}}\left[((\mu_i - \bar{\mu}_{-i}) + (\bar{\mu}_{-i} - \widehat{\bar{\mu}}_{-i}))^2\right]
$$
$$
= (\mu_i - \bar{\mu}_{-i})^2 + \mathrm{Var}_{\mathbf{Y}}[\widehat{\bar{\mu}}_{-i}]
$$
$$
= (\mu_i - \bar{\mu}_{-i})^2 + \frac{1}{(n-1)^2}\sum_{i'\neq i}\mathrm{Var}_{\mathbf{Y}}[\widehat{\mu}_{i'}]
$$
$$
= (\mu_i - \bar{\mu}_{-i})^2 + \frac{1}{(n-1)^2}\sum_{i'\neq i}\frac{1}{m-1}\sigma^2(x_{i'})
$$

Combining together, we have

$$\mathbb{E}_{\mathbf{x},\mathbf{Y}}[(\mu_i - b_i^j)^2]$$

$$=(1-\lambda_i^j)^2\mathbb{E}_{\mathbf{x}}\left[\frac{1}{m-1}\sigma^2(x_i)\right] + (\lambda_i^j)^2\mathbb{E}_{\mathbf{x}}\left[(\mu_i - \bar{\mu}_{-i})^2 + \frac{1}{(n-1)^2}\sum_{i'\neq i}\frac{1}{m-1}\sigma^2(x_{i'})\right]$$

$$=\frac{1}{m-1}\mathbb{E}_{x\sim\mathcal{D}}[\sigma^2(x)](1-\lambda_i^j)^2 + \frac{1}{(n-1)(m-1)}\mathbb{E}_{x\sim\mathcal{D}}[\sigma^2(x)](\lambda_i^j)^2 + \frac{n}{n-1}\mathrm{Var}_{x\sim\mathcal{D}}[\mu(x)](\lambda_i^j)^2$$

$$=\frac{n}{n-1}(s_2 + v_2)(\lambda_i^j)^2 - 2v_2\lambda_i^j + v_2.$$

Therefore, optimal $\lambda_i^j$ is

$$(\lambda_i^j)^* = \frac{n-1}{n}\frac{v_2}{s_2 + v_2}.$$

This holds for all $i, j$, so we complete the proof.