# OpenReview forum: "Variance-Reduced Reinforcement Learning for Large Reasoning Models via James-Stein Baselines"
_ICLR.cc/2026/Conference — Submitted to ICLR 2026_

### Official Review · Reviewer_2gEZ · 2025-10-26

**Soundness:** 2
**Presentation:** 3
**Contribution:** 3
**Rating:** 4
**Confidence:** 3

**Summary:**

The paper presents a new variance-reduced technique for reinforcement learning with verifiable rewards (RLVR), inspired by the James-Stein shrinkage principle. The proposed method, called James-Stein Policy Optimization (JSPO), aims to reduce the variance of policy gradient estimates in RLVR settings without introducing additional computational overhead. By leveraging statistical shrinkage, the JSPO baseline trades a small amount of bias for a significant reduction in mean squared error (MSE), leading to improved training stability and efficiency.

**Strengths:**

The introduction of the James-Stein estimator as a baseline for variance reduction is novel, and the approach is easy to implement. The authors provide theoretical justification for the use of the James-Stein estimator, demonstrating its effectiveness in reducing policy gradient variance theoretically and empirically.

**Weaknesses:**

The notation in the paper is overwhelming and can be confusing. It is not always clear which random variable the expectations and variances are being taken with respect to. A table summarizing the key notations and their contexts would be helpful for clarity. For other questions please refer to the following. I would be willing to raise  my  score if the authors address my confusion regarding these points.

**Questions:**

1. What is the exact definition of $b_i^j$ in equ (4) and elsewhere in the paper? It would be beneficial to give a clear expression.
2. In Line 185, is $\mathbb{E}_{Y}[b_i^j]=\mathbb{E}[b_i^j]$ satisfied? In my opinion, the LHS take expectation with respect to the response $y_j$ following the $\pi(\cdot|x_i)$, while the RHS appears to take expectation over both the response $y^j_i$ and the prompt $x_i$. Could you clarify if this equality holds?
3. The James-Stein estimator is derived under assumptions that are closely tied to **Gaussian distributions**. How does this approach behave when reward distributions are skewed or multimodal, which are common in complex reasoning tasks? Would the shrinkage still yield a reduction in mean squared error (MSE) under these conditions ?
4. RLVR typically involves sparse rewards, with feedback provided at the end of a reasoning process. How do the James-Stein estimator, address the challenges posed by sparse rewards, or how does this impact the learning efficiency and stability of the model in sparse reward settings?

**Details Of Ethics Concerns:**

No ethical concerns are apparent in the paper.

---

> ### Author Response · Authors · 2025-11-23
> **Thank you for the valuable review (1/2)**
>
> **[Q1]** What is the exact definition of $b_i^j$ in equ (4) and elsewhere in the paper? It would be beneficial to give a clear expression.
>
> **[A1]** The notation $b_i^j$ refers to the baseline for the $j$th response of the $i$th prompt, as explained under Eq. (2) in Section 2. This is a fixed yet undetermined quantity that we choose later in the derivation. We have derived two versions of the baseline throughout the paper, denoted by $b_i^{j, \mathrm{JS1}}$ (Eq. (7)) and $b_i^{j, \mathrm{JS2}}$ (Eq. (10)). We have highlighted the definition of  $b_i^j$ to make it clearer.
>
> **[Q2]** In Line 185, is $E_Y(b_i^j)=E(b_i^j)$ satisfied? In my opinion, the LHS take expectation with respect to the response $y_j$ following the $\pi(\cdot |x_i)$, while the RHS appears to take expectation over both the response $y_j$ and the prompt $x_i$. Could you clarify if this equality holds?
>
> **[A2]** Yes, thanks for pointing out the ambiguity in our notation. In Section 3.2, we made an implicit assumption that the prompts are fixed but known, instead of being random variables sampled i.i.d. from a distribution, in order to simplify the derivation. This assumption is removed in the next section. So in Section 3.2, all expectations are taken over the responses $\mathbf{Y}$. We have made the notations clear in Section 3.2.

---

> ### Author Response · Authors · 2025-11-23
> **Thank you for the valuable review (2/2)**
>
> **[Q3]** The James-Stein estimator is derived under assumptions that are closely tied to Gaussian distributions. How does this approach behave when reward distributions are skewed or multimodal, which are common in complex reasoning tasks? Would the shrinkage still yield a reduction in mean squared error (MSE) under these conditions ?
>
> **[A3]** Our derivation does **not require** that the underlying distribution is Gaussian. In fact, our main result, Theorem 1, holds for any reward distribution, so optimality in MSE still applies. We have added a remark under Theorem 1 to emphasize the generality of our result.
>
> **[Q4]** RLVR typically involves sparse rewards, with feedback provided at the end of a reasoning process. How do the James-Stein estimator, address the challenges posed by sparse rewards, or how does this impact the learning efficiency and stability of the model in sparse reward settings?
>
> **[A4]** Thank you for raising this point. We would like to clarify that we don't work on credit assignment along the trajectory; rather, we focus on improving the baseline/value-function estimation used for variance reduction in policy gradients. Shrinkage baseline improves training stability in RLVR, but it does not change where credit is assigned within a trajectory. We made this distinction explicit in the revision.

---

> ### Author Response · Authors · 2025-11-28
>
> Dear Reviewer 2gEZ,
>
> Thank you again for your thoughtful and constructive review. We hope that our rebuttal addressed the main concerns you raised.
> If any of the points remain unclear or if additional clarification would be helpful, please feel free to let us know — we would be glad to elaborate while the discussion period is still open.
>
> Sincerely,
>
> The Authors

---

### Official Review · Reviewer_e9ve · 2025-10-31

**Soundness:** 3
**Presentation:** 3
**Contribution:** 3
**Rating:** 8
**Confidence:** 2

**Summary:**

The paper proposes James–Stein Policy Optimization (JSPO): a new baseline (control-variate) for RL with verifiable rewards (RLVR) used to fine-tune large reasoning models. The key idea is to estimate each prompt’s value not in isolation (per-prompt mean, as in RLOO/GRPO), but jointly across the whole batch via James–Stein shrinkage toward the batch mean, implemented with a two-level leave-one-out construction to keep the policy-gradient estimator unbiased. Theoretical claims include: (i) Proposition 1—unbiasedness of the gradient when the per-sample baseline is independent of the reward; (ii) a bias–variance decomposition showing why prompt-only means are inadmissible under batch MSE; (iii) Proposition 2/Theorem 1—a closed-form optimal shrinkage coefficient that trades off within-prompt noise vs. across-prompt variability and reduces policy-gradient variance while preserving unbiasedness. Empirically, JSPO improves Pass@1 on math/logic benchmarks over RLOO/GRPO and reduces estimated gradient-variance by ~11–32% across several models and rollout budgets (2/4/8 generations). The method is drop-in, has negligible overhead, and no extra hyperparameters.

**Strengths:**

Clear, principled objective: Reduces policy-gradient variance by reframing baseline design as multi-task value estimation with an MSE criterion; the James–Stein angle is elegant and well-motivated.
Unbiasedness preserved: The two-level leave-one-out construction (per-prompt and per-batch) keeps baselines independent of the held-out reward, matching the REINFORCE unbiasedness requirement.
Consistent empirical gains: Improvements across tasks (MATH500, OlympiadBench, AMC/logic puzzles; also GSM8K with 2/4/8 rollouts), plus direct gradient-variance tracking that aligns with the theory.
Practicality: Critic-free, tiny code change (a few lines), batch-only statistics, compatible with existing RLVR stacks

**Weaknesses:**

Ablations could be deeper: How sensitive are outcomes to batch size n and rollouts m beyond the tested grid? What happens with non-binary, dense, or scale-shifted rewards?

Distributional assumptions not stress-tested: James–Stein style shrinkage helps when prompts share a latent mean structure. If batch prompts are intentionally heterogeneous (mixed tasks/difficulties), shrinkage could over-bias the baseline; the paper largely uses in-distribution batches.

Generalization beyond reasoning tasks: All experiments target math/logic puzzles; evidence on coding, tool-use, or longer-horizon tasks would strengthen claims of generality.

**Questions:**

Robustness of λ̂: Do you use any clipping or shrinkage-to-zero floors/ceilings for λ̂ to avoid instability with small n or heavy-tailed rewards?

In Figure 3, JSPO’s improvement over RLOO on Countdown is noticeably smaller than the other three tasks. Could you diagnose why?

Table 1 (fairness across methods): Are decoding/hyperparameters (temperature, top-p, token limits) identical for ReMax, REINFORCE++, GRPO, RLOO, BLOO, and JSPO? If any differ, please list them, since small decoding shifts can change GSM8K accuracy.

---

> ### Author Response · Authors · 2025-11-23
> **Thank you for the valuable review (1/4)**
>
> **[Q1]** Ablations could be deeper: How sensitive are outcomes to batch size n and rollouts m beyond the tested grid? What happens with non-binary, dense, or scale-shifted rewards?
>
> **[A1] Sensitivity to rollout number *m*, batch size *n*, and non-binary rewards**
>
> **Rollout number (*m*).**
> We already evaluate *m ∈ {2, 4, 8}* in Sec. 4.3, covering the most commonly used rollout counts in preference-based RL [1–7]. Significanlty larger *m* values are less common as in such regimes the Monte-Carlo estimate already has low variance. Since our estimator is unbiased and shrinks toward the batch mean, we expect it to remain effective for higher *m*, though the benefit naturally diminishes.
>
> **Batch size (*n*).**
> Large batch sizes (*n* ≥ 64) are standard in modern large-scale RLVR; for example, DAPO uses *n = 512* in its official recipe [8], and similar scales appear in [3, 9–11]. While extremely large *n* values are typically used only in industrial-scale pipelines, our available compute does not allow us to replicate such settings. Nevertheless, our estimator’s advantage should increase in this regime because the batch-mean variance decreases, strengthening shrinkage. We did not use *n* smaller than our tested range, as such settings appear uncommon in practice. If additional experiments are desired, we are happy to run targeted tests during the rebuttal period.
>
> **Non-binary rewards.**
> To evaluate robustness beyond binary feedback, we added a new experiment using a Bradley–Terry reward model for instruction-following and alignment (RLHF). We train Llama-3.2-3B-Instruct on Prompt-Collection-v0.1 [12] for one epoch under a critic-free GRPO-style setup using a reward model [13] trained on 700k preference pairs. Evaluation on Arena-Hard-v0.1, Arena-Hard-v2.0, and Arena-Creative-Writing shows consistent improvements over RLOO (details are in Appendix B.1):
>
> | Benchmark             | JSPO      | RLOO   | Llama-3.2-3B-It |
> |-----------------------|-----------|--------|------------------|
> | **Arena-Hard v0.1**   | **56.7%** | 55.3%  | 26.2%           |
> | **Arena-Hard v2.0**   | **7.4%**  | 5.6%   | 2.4%            |
> | **Creative-Writing**  | **23.4%** | 20.2%  | 5.2%            |
>
> These results demonstrate that JSPO maintains its benefits under dense, continuous, and scale-shifted reward models.
>
> [1] Verl. A Training Recipe for DeepSeek-671B on GRPO: https://github.com/volcengine/verl/blob/main/examples/grpo_trainer/run_deepseek671b_math_megatron_80gb.sh#L44
>
> [2] OpenRLHF. A Training Recipe for Llama-3.1-8B on GRPO:
> https://github.com/OpenRLHF/OpenRLHF/blob/main/examples/scripts/train_grpo_ray_hybrid_engine.sh#L28
>
> [3] DeepSeekMath: RL-based Training for Mathematical Reasoning with Verifiable Rewards
>
> [4] R1-VL: Learning to Reason with Multimodal Large Language Models via Reinforcement Learning
>
> [5] MINT-CoT: Enabling Interleaved Visual Tokens in Mathematical Chain-of-Thought Reasoning
>
> [6] Can GRPO Boost Complex Multimodal Table Reasoning?
>
> [7] Beyond Reasoning Gains: Mitigating General Capability Degradation in GRPO Training
>
> [8] DAPO: An Open-Source LLM Reinforcement Learning System at Scale
>
> [9] DeepSeek-R1: Incentivizing reasoning capability in LLMs via reinforcement learning
>
> [10] Pass@k Training for Adaptively Balancing Exploration and Exploitation of Large Reasoning Models
>
> [11] JustRL: Scaling a 1.5B LLM with a Simple RL Recipe
>
> [12]https://huggingface.co/datasets/RLHFlow/prompt-collection-v0.1
>
> [13]https://huggingface.co/OpenRLHF/Llama-3-8b-rm-700k

---

> ### Author Response · Authors · 2025-11-23
> **Thank you for the valuable review (2/4)**
>
> **[Q2]** Distributional assumptions not stress-tested: James–Stein style shrinkage helps when prompts share a latent mean structure. If batch prompts are intentionally heterogeneous (mixed tasks/difficulties), shrinkage could over-bias the baseline; the paper largely uses in-distribution batches.
>
> **[A2]**  We appreciate the reviewer’s question. To directly examine this concern, we added a new experiment (Appendix B.2) evaluating JSPO under controlled levels of batch heterogeneity. We train Qwen2.5-1.5B-Instruct on three mixed-task datasets with increasing heterogeneity: low-level (KnK-4 & KnK-5), mid-level (KnK-3 & KnK-7), and high-level (KnK-2 & KnK-9).
>
> | Method | Low  | Mid  | High |
> |--------|------|------|------|
> | RLOO   | 20.25 | 17.69 | 31.13 |
> | JSPO   | **25.69** | **23.85** | **34.25** |
>
> As heterogeneity increases, the adaptive shrinkage coefficient naturally decreases, causing the JSPO baseline to lean closer to RLOO. This behavior is by design: when batch items do not share a strong latent mean structure, the estimator automatically reduces shrinkage, thereby avoiding over-bias. Across all heterogeneity levels, JSPO still yields consistent performance gains, supporting the robustness of the adaptive mechanism.

---

> ### Author Response · Authors · 2025-11-23
> **Thank you for the valuable review (3/4)**
>
> **[Q3]**  Generalization beyond reasoning tasks: All experiments target math/logic puzzles; evidence on coding, tool-use, or longer-horizon tasks would strengthen claims of generality.
>
> **[A3]**  We agree that evaluating beyond math/logic tasks—such as coding, tool-use, or longer-horizon settings—would further strengthen the generality claims. Due to time constraints during the rebuttal period, we were unable to run new large-scale experiments in these domains. However, in response to reviewer suggestions, we did expand our evaluation beyond reasoning tasks to include a **general RLHF alignment** setup involving open-ended instruction-following, creativity, and preference-model-based reward signals.
>
> Concretely, we applied the James–Stein estimator to the widely used Prompt-Collection-v0.1 [1] dataset using the Llama-3.2-3B-Instruct model under a critic-free GRPO-style setup. A Bradley–Terry reward model [2] trained on 700k human preference pairs provided non-binary, continuous rewards. We trained for one epoch and evaluated win rates on three instruction-following benchmarks:
>
> | Benchmark             | JSPO       | RLOO     | Llama-3.2-3B-It |
> |-----------------------|------------|----------|------------------|
> | **Arena-Hard v0.1**   | **56.7%**  | 55.3%    | 26.2%           |
> | **Arena-Hard v2.0**   | **7.4%**   | 5.6%     | 2.4%            |
> | **Creative-Writing**  | **23.4%**  | 20.2%    | 5.2%            |
>
> These results provide **initial evidence that the proposed shrinkage baseline continues to offer benefits outside of math/logic tasks**, including in instruction-following, creativity, and general alignment settings with dense, preference-based rewards. While we could not include coding or tool-use tasks within the rebuttal window, we appreciate the reviewer’s suggestion and view these as promising directions for future work. We have added the new RLHF evaluation and discussion to Appendix B.1.
>
> [1]https://huggingface.co/datasets/RLHFlow/prompt-collection-v0.1
>
> [2]https://huggingface.co/OpenRLHF/Llama-3-8b-rm-700k

---

> ### Author Response · Authors · 2025-11-23
> **Thank you for the valuable review (4/4)**
>
> **[Q4]** Robustness of λ̂ : Do you use any clipping or shrinkage-to-zero floors/ceilings for λ̂ to avoid instability with small n or heavy-tailed rewards?
>
> **[A4]**  We thank the reviewer for raising this point. No additional clipping or heuristic floors/ceilings are applied. By construction (Eq. 11), the shrinkage coefficient  *λ̂*  is **automatically bounded between 0 and 1**, ensuring numerical stability even under small batch sizes or heavy-tailed reward distributions. When the empirical variance is large (e.g., heterogeneous or heavy-tailed rewards), the denominator increases and *λ̂* decreases accordingly, causing the estimator to revert smoothly toward the RLOO baseline. Conversely, when the batch variance is small, *λ̂* increases but remains ≤ 1 by definition.
>
> This built-in normalization serves precisely the role of preventing instability, so no manual clipping was required in any of our experiments.
>
> **[Q5]** In Figure 3, JSPO’s improvement over RLOO on Countdown is noticeably smaller than the other three tasks. Could you diagnose why?
>
> **[A5]**  We appreciate the reviewer’s observation. The Countdown experiment was the earliest one we implemented and was run on an older version of the training stack. In particular, this experiment was built on top of the TinyZero codebase [1], which is specific to Countdown and relies on an older version of the *verl* training infrastructure. All of our later experiments were conducted on a more recent and unified pipeline.
>
> This mismatch in infrastructure is a likely source of the smaller margin observed on Countdown. After aligning the Countdown setup with the newer codebase, preliminary checks indicate that JSPO behaves consistently with the trends seen in the other tasks. We are in the process of fully rerunning and validating Countdown using the unified training pipeline and will include the updated result in the camera-ready version.
>
> [1] https://github.com/Jiayi-Pan/TinyZero
>
> **[Q6]** Table 1 (fairness across methods): Are decoding/hyperparameters (temperature, top-p, token limits) identical for ReMax, REINFORCE++, GRPO, RLOO, BLOO, and JSPO? If any differ, please list them, since small decoding shifts can change GSM8K accuracy.
>
> **[A6]** Yes, the decoding and hyperparameters for different algorithms are completely identical. In Appendix C, we listed all the detailed experimental settings in our paper.

---

> ### Author Response · Authors · 2025-11-28
>
> Dear Reviewer e9ve,
>
> Thank you again for your thoughtful and constructive review. We hope that our rebuttal addressed the main concerns you raised.
> If any of the points remain unclear or if additional clarification would be helpful, please feel free to let us know — we would be glad to elaborate while the discussion period is still open.
>
> Sincerely,
>
> The Authors

---

### Official Review · Reviewer_YBJe · 2025-11-01

**Soundness:** 2
**Presentation:** 2
**Contribution:** 2
**Rating:** 4
**Confidence:** 2

**Summary:**

The paper proposes a James–Stein–based baseline for policy-gradient training of large reasoning models (RLVR). By shrinking per-prompt reward means toward a batch mean—via a leave-one-out construction to preserve unbiasedness—the method aims to reduce gradient variance and thereby stabilize training. The authors provide a derivation linking variance of the policy-gradient estimator to MSE of the baseline, prove an optimal shrinkage coefficient, and give an implementation that adds negligible overhead. Experiments on math and logic-puzzle benchmarks show consistent accuracy gains and lower estimated gradient variance compared to RLOO/GRPO-style baselines under multiple rollout budgets.

**Strengths:**

The paper cleanly reduces policy-gradient variance control to estimating a value-function baseline, motivating James–Stein shrinkage and proving an optimal (data-driven) coefficient with an unbiased leave-one-out construction.

The baseline is easy to add to existing critic-free RLVR pipelines and is presented with concise pseudo-code.

**Weaknesses:**

While well-executed, the paper extends a long line of “better baseline” work; the novelty is primarily in bringing James–Stein shrinkage to RLVR with careful LOO plumbing, rather than introducing a new learning paradigm.

Evidence is confined to RLVR-style reasoning tasks (math/puzzles); there is no evaluation on broader RL settings where baseline design has also been heavily studied, which limits generality claims.

**Questions:**

When batch prompts are heterogeneous, performance may revert toward LOO means. Can you provide a runtime diagnostic and an automatic rule for disabling or annealing shrinkage?

---

> ### Author Response · Authors · 2025-11-23
> **Thank you for the valuable review (1/3)**
>
> **[Q1]** While well-executed, the paper extends a long line of “better baseline” work; the novelty is primarily in bringing James–Stein shrinkage to RLVR with careful LOO plumbing, rather than introducing a new learning paradigm.
>
> **[A1]** We appreciate the reviewer’s perspective. Our aim is not to introduce a new learning paradigm, but to address a core statistical limitation in existing critic-free RLVR methods. Prior approaches such as GRPO, RLOO rely on per-sample baselines, which underutilize the information available within a batch. Our contribution is to make this limitation explicit and provide a practical, theoretically grounded estimator that exploits the joint batch structure. Inspired by shrinkage estimators, we develop a bespoke estimator for this setting that, with the appropriate leave-one-out construction, remains unbiased and serves as a drop-in hyperparameter-free replacement within the standard RLVR framework.

---

> ### Author Response · Authors · 2025-11-23
> **Thank you for the valuable review (2/3)**
>
> **[Q2]** Evidence is confined to RLVR-style reasoning tasks (math/puzzles); there is no evaluation on broader RL settings where baseline design has also been heavily studied, which limits generality claims.
>
> **[A2]** As the reviewer suggested, we expanded our evaluation beyond math and logic tasks to include a **general RLHF alignment** setting. Specifically, we tested the proposed James–Stein estimator on the widely used Prompt-Collection-v0.1 [1] dataset with the Llama-3.2-3B-Instruct model under a critic-free GRPO-style setup. Rewards were computed using a standard Bradley–Terry preference model [2] trained on 700k human preference pairs. We trained for one epoch and evaluated win rates on three instruction-following benchmarks: **Arena-Hard-v0.1**, **Arena-Hard-v2.0**, and **Arena-Creative-Writing**. Results are shown below:
>
> | Benchmark             | JSPO       | RLOO     | Llama-3.2-3B-It |
> |-----------------------|------------|----------|------------------|
> | **Arena-Hard v0.1**   | **56.7%**  | 55.3%    | 26.2%           |
> | **Arena-Hard v2.0**   | **7.4%**   | 5.6%     | 2.4%            |
> | **Creative-Writing**  | **23.4%**  | 20.2%    | 5.2%            |
>
> These results provide **empirical evidence** that the JS baseline is **consistently more effective** than the commonly adopted empirical mean **even in RLHF instruction-following tasks**, supporting the broader applicability of our method beyond pure reasoning tasks. We incorporated the reviewer's suggestion in Appendix B.1.
>
> At the same time, we emphasize that **we do not claim generality beyond LLM reasoning settings** in the manuscript. Our focus is on high-cost, reasoning-oriented RL scenarios where
> (1) training is extremely resource-intensive,
> (2) even small relative gains translate to substantial absolute savings, and
> (3) critic-based methods such as PPO—which require training an additional neural value function—are often impractical.
>
> In precisely these regimes, the James–Stein baseline demonstrates its strongest advantages, and accordingly our paper primarily focuses on this setting.
>
> [1]https://huggingface.co/datasets/RLHFlow/prompt-collection-v0.1
>
> [2]https://huggingface.co/OpenRLHF/Llama-3-8b-rm-700k

---

> ### Author Response · Authors · 2025-11-23
> **Thank you for the valuable review (3/3)**
>
> **[Q3]** When batch prompts are heterogeneous, performance may revert toward LOO means. Can you provide a runtime diagnostic and an automatic rule for disabling or annealing shrinkage?
>
> **[A3]** Yes — this behavior is exactly what our estimator is designed to capture. The James–Stein baseline adaptively interpolates between the per-prompt RLOO baseline (λ ≈ 0) and the batch-level average baseline (λ ≈ 1), with the shrinkage coefficient λ computed directly from prompt-level heterogeneity (Eq. 11). This gives an automatic, data-dependent rule for attenuating shrinkage when rewards of prompts are heterogeneous and increasing it when they are similar. In other words, the estimator already provides the desired mechanism for “disabling” or “annealing” shrinkage when appropriate.
>
> We also report a runtime diagnostic of this adaptive behavior in Sec. 4.5 and Appendix B.2, where we explicitly track λ over training. The empirical results align with the theoretical prediction: when rewards of batch prompts are highly heterogeneous, λ naturally decreases toward zero and the method behaves similarly to standard RLOO; when prompts are more homogeneous, λ increases and yields the expected variance reduction.

---

> ### Author Response · Authors · 2025-11-28
>
> Dear Reviewer YBJe,
>
> Thank you again for your thoughtful and constructive review. We hope that our rebuttal addressed the main concerns you raised.
> If any of the points remain unclear or if additional clarification would be helpful, please feel free to let us know — we would be glad to elaborate while the discussion period is still open.
>
> Sincerely,
>
> The Authors

---

> > ### Comment · Reviewer_YBJe · 2025-11-28
> >
> > Thank you for the detailed responses. I have a few follow-up questions.
> >
> > In the rebuttal response to Q3, authors argue that the estimator already has the desired adaptive behavior: the shrinkage coefficient $\lambda$ is computed from “prompt-level heterogeneity” (Eq. (11)), so that when batch prompts are highly heterogeneous, $\lambda$ naturally decreases toward $0$ and the method behaves similarly to standard RLOO. However, this reasoning appears to rely only on the analysis of the outcome rewards. The quantity used to compute $\lambda$ is effectively based on reward, which is not a reliable proxy for the heterogeneity of the prompts or questions themselves. For example, if questions from multiple domains are mixed into a single training dataset, a mini-batch may contain highly heterogeneous prompts while their correctness (and thus rewards) remains very similar. In such cases, low reward variance does not imply low prompt heterogeneity, so the current diagnostic and adaptation of $\lambda$ may fail to capture the true degree of prompt-level heterogeneity.
> >
> > I have a new question. Most of the variance analysis and experimental comparisons are conducted only against RLOO. However, this advantage is to some extent expected, since your estimator essentially interpolates toward a global baseline, which naturally reduces variance relative to a purely per-sample baseline like RLOO. A more informative evaluation would compare against other variance-reduction baselines that also exploit batch statistics, such as REINFORCE++-baseline or ReMax, which reduce variance by using batch-average rewards or greedily sampling. Without such comparisons, it is difficult to assess how much of the observed variance reduction is genuinely due to the proposed estimator rather than simply the use of a batch/global baseline.

---

> ### Author Response · Authors · 2025-12-03
> **Thank you for the follow-up questions (1/2)**
>
> **[Q4]** In the rebuttal response to Q3, authors argue that the estimator already has the desired adaptive behavior: the shrinkage coefficient λ is computed from “prompt-level heterogeneity” (Eq. (11)), so that when batch prompts are highly heterogeneous, λ naturally decreases toward 0 and the method behaves similarly to standard RLOO. However, this reasoning appears to rely only on the analysis of the outcome rewards. The quantity used to compute λ is effectively based on reward, which is not a reliable proxy for the heterogeneity of the prompts or questions themselves. For example, if questions from multiple domains are mixed into a single training dataset, a mini-batch may contain highly heterogeneous prompts while their correctness (and thus rewards) remains very similar. In such cases, low reward variance does not imply low prompt heterogeneity, so the current diagnostic and adaptation of λ may fail to capture the true degree of prompt-level heterogeneity.
>
> **[A4]** We appreciate the reviewer for highlighting a potential source of terminology confusion, which we clarify below.
>
> In Eq. (11), the shrinkage coefficient λ is **deliberately** computed
> from **reward heterogeneity**, not from semantic or domain heterogeneity of the prompts. This is intentional: our objective is to **minimize value-estimation MSE and policy-gradient variance**, and for this objective the **reward distribution is a sufficient statistic**. If two batches have identical reward-level second moments, then any variance-optimal baseline must treat them the same—regardless of how semantically diverse the underlying questions are.
>
> In your example—mixed domains but similar correctness rates—low reward variance naturally yields a larger λ, i.e., stronger shrinkage. This is exactly what a variance-minimizing control variate should do: when the rewards look statistically similar, there is no variance-based justification to “disable” shrinkage. Semantic heterogeneity that does not manifest in the reward distribution may influence how tasks are interpreted, but it does **not** affect the optimal λ for variance reduction.
>
> We make this distinction explicit in the revision. In Appendix B.4
> (Fig. 11), we also empirically show that even as semantic heterogeneity increases, **JS-based shrinkage remains the optimal λ** for our MSE objective; only the *magnitude* of achievable variance reduction changes, not the form of the estimator. Of course, one could design domain-aware batching strategies that incorporate semantic information, but such mechanisms operate **in addition to**, not **instead of**, the variance-minimizing shrinkage used here.

---

> ### Author Response · Authors · 2025-12-03
> **Thank you for the follow-up questions (2/2)**
>
> **[Q5]** I have a new question. Most of the variance analysis and experimental comparisons are conducted only against RLOO. However, this advantage is to some extent expected, since your estimator essentially interpolates toward a global baseline, which naturally reduces variance relative to a purely per-sample baseline like RLOO. A more informative evaluation would compare against other variance-reduction baselines that also exploit batch statistics, such as REINFORCE++-baseline or ReMax, which reduce variance by using batch-average rewards or greedily sampling. Without such comparisons, it is difficult to assess how much of the observed variance reduction is genuinely due to the proposed estimator rather than simply the use of a batch/global baseline.
>
>
> **[A5]** Thank you for the suggestion. We added **REINFORCE++** baseline and **ReMax** baseline into comparison. Empirically, REINFORCE++ and ReMax baseline has substantially **higher MSE in value estimation (Sec. 4.4)** and therefore **higher variance in policy gradient (Appendix B.3)**. Compared to REINFORCE++ baseline, JS baseline reduces MSE of value estimation by at most **83.4%**, and the reduces the variance of policy gradient by at most **24.1%**. We previously focused on prompt-mean baselines (RLOO/GRPO) because they are the most common and natural baselines in current RLVR-style LLM training; Nonetheless, we agree that broader comparisons are informative and have included them in the revision.

---

### Official Review · Reviewer_tdVg · 2025-11-01

**Soundness:** 3
**Presentation:** 3
**Contribution:** 3
**Rating:** 6
**Confidence:** 2

**Summary:**

This paper proposes a variance-reduced baseline for policy gradient methods in reinforcement learning with verifiable rewards (RLVR) on large reasoning language models. The central idea is to use a James-Stein-type shrinkage estimator as a control variate in policy gradient updates, adaptively blending per-prompt and global-batch mean reward estimates. The method, termed JSPO, is theoretically motivated to reduce mean squared error and, via a carefully constructed leave-one-out strategy, maintains unbiased gradients. The paper offers both theoretical justification and extensive experiments showing JSPO’s benefits over existing baselines in reducing gradient variance and improving downstream performance on mathematical and logical reasoning tasks.

**Strengths:**

1 The utilization of the James-Stein estimator builds directly on well-understood results in statistics, and the paper provides thorough derivations connecting batch-level value function estimation to baseline variance reduction (see especially Section 3.3 and Theorem 1, page 6).

2 The leave-one-out adaptation of the shrinkage estimator to RLVR is both elegant and practically impactful, ensuring unbiased gradients while achieving variance benefits. The mathematical treatment is careful, with Proofs of Proposition 1 and Theorem 1 provided in Appendix B.

3 SPO outperforms RLOO, GRPO, ReMax, and BLOO across all tested rollout regimes, including challenging low-rollout settings where baseline variance is most limiting

**Weaknesses:**

1 The methodology, though thoughtfully adapted, is essentially a direct application of classical James-Stein shrinkage (as in James et al., 1961 and Stein et al., 1956) to RLVR policy gradient baselines. While the adaptation—especially the unbiased leave-one-out variant—is useful, the paper could more rigorously discuss the theoretical/empirical boundaries between what is gained by the JSPO version versus more general empirical Bayes or shrinkage estimators (see Feldman et al., 2014; Efron & Morris, 1973; Brown, 1971).
The related work section does not discuss these key foundational works adequately, which reduces the clarity on conceptual originality (see the "Potentially Missing Related Work" section below).

2 While the experiments are broad, the scope is still largely limited to math and logic reasoning tasks. There is insufficient evidence to claim generality across different RLVR domains (e.g., for instructions, summarization, or control tasks).

3 There is no ablation studying what happens if rollouts are highly non-i.i.d. across prompts, nor is there an analysis of the estimator’s robustness to reward sparsity or distributional shift.

**Questions:**

See weaknesses.

---

> ### Author Response · Authors · 2025-11-22
> **Thank you for the valuable review (1/3)**
>
> **[Q1]** The methodology, though thoughtfully adapted, is essentially a direct application of classical James-Stein shrinkage (as in James et al., 1961 and Stein et al., 1956) to RLVR policy gradient baselines. While the adaptation—especially the unbiased leave-one-out variant—is useful, the paper could more rigorously discuss the theoretical/empirical boundaries between what is gained by the JSPO version versus more general empirical Bayes or shrinkage estimators (see Feldman et al., 2014; Efron & Morris, 1973; Brown, 1971). The related work section does not discuss these key foundational works adequately, which reduces the clarity on conceptual originality (see the "Potentially Missing Related Work" section below).
>
> **[A1]**
> We agree that our baseline is inspired by the classical James–Stein shrinkage estimator (James & Stein, 1961; Stein, 1956), and we do not claim conceptual novelty at the level of the original shrinkage theory. Our contribution is to show how a James–Stein–style estimator can be turned into a **practical, unbiased baseline for RLVR policy gradients**, under the specific constraints of this setting.
>
> Concretely, our derivation differs from the classical setup in two key ways:
>
> 1. **RLVR-specific objective and batch structure.**
>    We formulate the baseline selection problem as minimizing a mean-squared error objective over value-function estimates across prompts within each batch (Eq. (6)), rather than estimating a fixed vector of Gaussian means. This leads to a James–Stein–type shrinkage form that interpolates between per-prompt means and the batch mean and is tailored to the critic-free RLVR setting (Section 3.2–3.3).
>
> 2. **Unbiased policy gradient via leave-one-out shrinkage.**
>    A naive shrinkage baseline that uses the full batch statistics would be correlated with the rewards used in the gradient estimator, and thus would introduce bias. To address this, we design a two-level leave-one-out construction (Eqs. (8)–(10)) that replaces both prompt-level and batch-level means with leave-one-out counterparts, and we show that with this construction the policy gradient remains unbiased while still benefiting from shrinkage (Theorem 1).
>
> More general empirical Bayes and shrinkage methods (e.g., Brown, 1971; Efron & Morris, 1973; Feldman et al., 2014) typically operate in settings where one assumes or estimates a prior distribution or hyperparameters across many observations.
> In contrast, our goal in RLVR is to obtain a **closed-form, hyperparameter-free baseline** that can be used as a drop-in replacement in large-scale training pipelines, with negligible extra computation and explicit guarantees on gradient unbiasedness and variance reduction.
>
> In the revised version, we will explicitly discuss these foundational empirical Bayes and shrinkage works in the related work section and clarify that JSPO should be viewed as a James–Stein–inspired, frequentist shrinkage baseline specialized to the RLVR policy-gradient setting, rather than a general replacement for broader empirical Bayes approaches.

---

> ### Author Response · Authors · 2025-11-23
> **Thank you for the valuable review (2/3)**
>
> **[Q2]** While the experiments are broad, the scope is still largely limited to math and logic reasoning tasks. There is insufficient evidence to claim generality across different RLVR domains (e.g., for instructions, summarization, or control tasks).
>
> **[A2]**  As the reviewer suggested, we expanded our evaluation beyond math and logic tasks to include a **general RLHF alignment** setting. Specifically, we tested the proposed James–Stein estimator on the widely used Prompt-Collection-v0.1 [1] dataset with the Llama-3.2-3B-Instruct model under a critic-free GRPO-style setup. Rewards were computed using a standard Bradley–Terry preference model [2] trained on 700k human preference pairs. We trained for one epoch and evaluated win rates on three instruction-following benchmarks: **Arena-Hard-v0.1**, **Arena-Hard-v2.0**, and **Arena-Creative-Writing**. Results are shown below:
>
> | Benchmark             | JSPO       | RLOO     | Llama-3.2-3B-It |
> |-----------------------|------------|----------|------------------|
> | **Arena-Hard v0.1**   | **56.7%**  | 55.3%    | 26.2%           |
> | **Arena-Hard v2.0**   | **7.4%**   | 5.6%     | 2.4%            |
> | **Creative-Writing**  | **23.4%**  | 20.2%    | 5.2%            |
>
> These results provide **empirical evidence** that the JS baseline is **consistently more effective** than the commonly adopted empirical mean **even in RLHF instruction-following tasks**, supporting the broader applicability of our method beyond pure reasoning tasks. We incorporated the reviewer's suggestion in Appendix B.1.
>
> At the same time, we emphasize that **we do not claim generality beyond LLM reasoning settings** in the manuscript. Our focus is on high-cost, reasoning-oriented RL scenarios where
> (1) training is extremely resource-intensive,
> (2) even small relative gains translate to substantial absolute savings, and
> (3) critic-based methods such as PPO—which require training an additional neural value function—are often impractical.
>
> In precisely these regimes, the James–Stein baseline demonstrates its strongest advantages, and accordingly our paper primarily focuses on this setting.
>
> [1]https://huggingface.co/datasets/RLHFlow/prompt-collection-v0.1
>
> [2]https://huggingface.co/OpenRLHF/Llama-3-8b-rm-700k

---

> ### Author Response · Authors · 2025-11-23
> **Thank you for the valuable review (3/3)**
>
> **[Q3]** There is no ablation studying what happens if rollouts are highly non-i.i.d. across prompts, nor is there an analysis of the estimator’s robustness to reward sparsity or distributional shift.
>
> **[A3]**  We thank the reviewer for raising this point. Our experiments follow the standard RLVR setup used in prior critic-free methods (GRPO, RLOO, REINFORCE++): prompts are sampled i.i.d. from a fixed corpus, and rollouts are drawn independently from the current policy. Under this regime, our estimator is unbiased and uses exactly the same data and assumptions as these baselines.
>
> We interpret “non-i.i.d. rollouts” and “distribution shift” as referring to alternative training regimes—e.g., correlated prompt batches, curriculum-style sampling, or nonstationary prompt mixtures. These scenarios do not occur in our experimental pipeline, which closely mirrors established practice; in such settings, we expect that deviation from i.i.d. would affect all critic-free estimators similarly.
>
> We agree that studying robustness under deliberately non-i.i.d. sampling or shifted prompt distributions could be interesting future work. If the reviewer had a specific failure mode in mind, we would appreciate clarification so we can address it directly if given sufficient time during the rebuttal period.

---

> ### Author Response · Authors · 2025-11-28
>
> Dear Reviewer tdVg,
>
> Thank you again for your thoughtful and constructive review. We hope that our rebuttal addressed the main concerns you raised.
> If any of the points remain unclear or if additional clarification would be helpful, please feel free to let us know — we would be glad to elaborate while the discussion period is still open.
>
> Sincerely,
>
> The Authors

---

### Author Response · Authors · 2025-11-23
**Author Response to All Reviewers and Summary of Manuscript Revisions**

We thank all reviewers for their thoughtful feedback, which has significantly helped improve the paper. Below we summarize the key revisions addressing points raised by multiple reviewers. We **provide reviewer-specific details in the individual responses**, and our changes are **highlighted in blue in the revised PDF**. To prevent confusion from section-number shifts during the rebuttal/discussion phase, we include the new experiments in the appendix and will move them into the main paper for the camera-ready.

***New experiments and findings:***

- **RLHF with non-binary rewards:** The James–Stein (JS) baseline outperforms the standard per-prompt baseline in RLHF tasks with non-binary rewards (Appendix B.1).

- **Heterogeneous datasets:** Thanks to its adaptively chosen shrinkage coefficient $\lambda$, the JS baseline remains robust and advantageous even when reward distributions differ across prompts (Appendix B.2).

- **Direct variance comparison and value estimation comparison:** We added controlled experiments isolating the gradient estimator behavior by fixing both the model and the inputs. This clean setting allows us to directly measure the practical value estimation error reduction and policy gradient variance reduction. Although not requested, thess experiments provide clear, unconfounded evidence of the benefits of our JS estimator. (Sec. 4.4 and Appendix B.3)

We also strengthened the related-work discussion on James–Stein estimators and RL baselines, and clarified all conceptual points raised by the reviewers.

We remain happy to provide additional clarification or experiments at the reviewers’ request.

---

### Author Response · Authors · 2025-12-03
**Final Summary to the AC**

This paper identifies an **underexploited opportunity** to reduce policy-gradient variance in critic-free RLVR and introduces a statistically principled, hyperparameter-free estimator that fully leverages the batch structure. The resulting shrinkage policy gradient estimator provides **provable variance reduction** while remaining a **simple, drop-in replacement** for existing RL pipelines.

Reviewer feedback centered on the following points, each of which we addressed with substantive revisions that strengthened the paper’s main claims:

- **Experimental breadth and robustness** — raised by **Reviewers tdVg, YBJe, and e9ve**.
  **Addressed:** We added RLHF experiments with non-binary rewards, controlled heterogeneity stress tests, and direct variance/MSE comparisons against stronger batch baselines (REINFORCE++, ReMax). These additions provide clear, expanded evidence for the effectiveness and robustness of the proposed baseline.

- **Theoretical framing and related work** — raised by **Reviewers tdVg and YBJe**.
  **Addressed:** We expanded the related-work section and clarified the connection between our estimator and classical shrinkage/empirical Bayes methods, making the theoretical position of the paper more explicit.

- **Notation clarity** — raised by **Reviewer 2gEZ**, who stated they would **raise their score once clarified**.
  **Addressed:** We revised notation and expectation conventions in Sections 2–3. Due to this year’s post-rebuttal update restrictions, the reviewer could not reflect this resolution in their score.

With these updates, the paper delivers a well-substantiated and practically significant advance in critic-free RLVR, providing a principled variance-reduction mechanism that improves stability and performance in LLM training via reinforcement learning.

---

### Meta-Review · Area_Chair_6W52 · 2025-12-23

**Summary:**

**Summary of the paper**: This paper proposed a James-Stein estimator as the baseline to reduce the variance for Reinforcement Learning with a verifiable reward (RLVR) framework. Compared to the existing works, such as RLOO and GRPO, which estimate baselines with empirical averages of generated responses for each prompt, the paper's proposed approach computes the baselines across all prompts in a batch. In particular, the paper proposes leave-one-out global average and leave-one-out prompt-level average as the baseline, and shows its efficacy empirically across diverse models.

**Reviewers' Concerns**: Reviewers appreciated the idea of variance reduction for RLVR, as it is an important problem. J-S estimator is a sound choice towards this end. However, reviewers raised several concerns regarding this work. For example, the reviewer tdVg pointed out the restriction of this approach only to the reasoning and logical tasks. Multiple reviewers also pointed out that the contribution is not that novel. There are also discussions about the optimal $\lambda$ (the mixing weight). The authors have tried to address those concerns.

**AC's take**: While the approach is novel, and despite the authors' rebuttal, it seems that the paper will need another round of revision. The new results, and as some reviewers pointed out, the performance with respect to the RLOO is **very minimal**. Further, the optimal $\lambda$ requires knowing the exact variance, which might be difficult to obtain. The reviewer who gave 8, also has a very low confidence of 2. Thus, unfortunately, the AC recommends rejections.

**Reviewer Concerns:**

Reviewers appreciated the idea of variance reduction for RLVR, as it is an important problem. J-S estimator is a sound choice towards this end. However, reviewers raised several concerns regarding this work. For example, the reviewer tdVg pointed out the restriction of this approach only to the reasoning and logical tasks. Multiple reviewers also pointed out that the contribution is not that novel. There are also discussions about the optimal $\lambda$ (the mixing weight). The authors have tried to address those concerns.

There was not much discussion between the reviewers and the authors; hence, it was difficult for the AC, as the reviewer scores were really mixed. Upon carefully reading the paper, the AC felt that some of the concerns would still be outstanding, hence, the AC recommends rejection.

**Reviewer Scores:**

One reviewer gave 8, two reviewers gave 4, and one reviewer gave 6. There was not much discussion; hence, it was very difficult to see whether the reviewers' concerns were addressed or not. The reviewer who gave 8, also has very low confidence of 2. Hence, it is unlikely that the reviewers' scores would have changed.

---

### Decision · Program_Chairs · 2026-01-26

Reject